

# Interannual Variability of Ammonia Concentrations over the United States: Sources and Implications

Luke D. Schiferl[1], Colette L. Heald[1,2], Martin Van Damme[3], Lieven Clarisse[3], Cathy Clerbaux[3,4], Pierre-François Coheur[3], John B. Nowak[5], J. Andrew Neuman[6,7], Scott C. Herndon[5], Joseph R. Roscioli[5], and Scott J. Eilerman[6,7]

[1]Department of Civil and Environmental Engineering, Massachusetts Institute of Technology, Cambridge, Massachusetts, USA
[2]Department of Earth, Atmospheric and Planetary Sciences, Massachusetts Institute of Technology, Cambridge, Massachusetts, USA
[3]Spectroscopie de l'atmosphère, Chimie Quantique et Photophysique, Université Libre de Bruxelles, Brussels (ULB), Belgium
[4]LATMOS/IPSL, UPMC Univ. Paris 06 Sorbonne Universités, UVSQ, CNRS, Paris, France
[5]Aerodyne Research, Inc., Billerica, Massachusetts, USA
[6]Cooperative Institute for Research in Environmental Sciences, University of Colorado Boulder, Boulder, Colorado, USA
[7]Chemical Sciences Division, Earth System Research Laboratory, NOAA, Boulder, Colorado, USA

*Correspondence to*: Luke D. Schiferl (schiferl@mit.edu)

**Abstract.** The variability of atmospheric ammonia ($NH_3$), emitted largely from agricultural sources, is an important factor when considering how inorganic fine particulate matter ($PM_{2.5}$) concentrations and nitrogen cycling are changing over the United States. This study combines new observations of ammonia concentration from the surface, aboard aircraft, and retrieved by satellite to both evaluate the simulation of ammonia in a chemical transport model (GEOS-Chem) and identify which processes control the variability of these concentrations over a 5-year period (2008–2012). We find that the model generally underrepresents the ammonia concentration near large source regions and fails to reproduce the extent of interannual variability observed at the surface during the summer (JJA). Variability in the base simulation surface ammonia concentration is dominated by meteorology (64 %) as compared to reductions in $SO_2$ and $NO_x$ emissions imposed by regulation (32 %) over this period. Introduction of year-to-year varying ammonia emissions based on animal population, fertilizer application, and meteorologically driven volatilization does not substantially improve the model comparison with observed ammonia concentrations, and these ammonia emissions changes have little effect on the simulated ammonia concentration variability compared to those caused by the variability of meteorology and acid-precursor emissions. There is also little effect on the $PM_{2.5}$ concentration due to ammonia emissions variability in the summer when gas-phase changes are favored, but variability in wintertime emissions, as well as in early spring and late fall, will have a larger impact on $PM_{2.5}$ formation. Further, this work highlights the need for continued improvement in both satellite-based and in situ ammonia measurements to better constrain the magnitude and impacts of spatial and temporal variability in ammonia concentrations.





## 1 Introduction

The modern agricultural system developed to feed an increasing human population relies heavily on artificially produced reactive nitrogen in the form of ammonia ($NH_3$). The intensification of agricultural practices has significantly perturbed the global nitrogen cycle over the past century, including increases in ammonia emissions to the atmosphere (Galloway and Cowling, 2002; Erisman et al., 2008; Sutton et al., 2008). Agricultural ammonia emissions contribute to inorganic fine particulate matter ($PM_{2.5}$) formation (e.g., ammonium sulfate and ammonium nitrate) in the atmosphere (Seinfeld and Pandis, 2006). $PM_{2.5}$ has numerous negative effects on human health, including respiratory and cardiovascular distress and an overall decrease in life expectancy (Pope et al., 2009). Agriculture has a large impact on $PM_{2.5}$ throughout the world, contributing up to 40 % of premature mortality due to outdoor air pollution in parts of Europe (Lelieveld et al., 2015). In the United States (US), ammonia emissions from agriculture exports alone react to increase population-weighted $PM_{2.5}$ concentration domestically by 0.36 µg m$^{-3}$, with contributions greater than 1 µg m$^{-3}$ in parts of the Midwest (Paulot and Jacob, 2014). Thus, the regulation of ammonia emissions may have the potential to reduce $PM_{2.5}$ in ammonia-limited areas (Pinder et al., 2006); and in a $SO_x$-limited environment ($SO_x$ = sulfur dioxide ($SO_2$) + sulfate ($SO_4^{2-}$)), ammonia can play a more important role leading to ammonium nitrate formation. However, this potential for ammonia emissions reductions to reduce $PM_{2.5}$ may be decreasing as $SO_2$ and $NO_x$ (nitric oxide (NO) + nitrogen dioxide ($NO_2$)) regulation are implemented in the US (Holt et al., 2015). $PM_{2.5}$ also contributes to reduced environmental visibility and affects the radiative budget of the earth (IPCC, 2013). Finally, the release of excess nitrogen from agricultural sources into the atmosphere will also increase nitrogen deposition fluxes, which can cause negative ecosystem effects such as acidification and eutrophication (Erisman et al., 2007). This is of particular concern in sensitive ecosystems such as alpine terrain and wetlands (Beem et al., 2010; Ellis et al., 2013).

The magnitude and timing of the ammonia emissions from agriculture is generally less well understood than for other anthropogenic emissions (e.g., mobile sources of $NO_x$, power plant emissions of $SO_x$). A sticky gas, ammonia is difficult to measure in situ, and this can lead to a low bias in measured concentrations (von Bobrutzki et al., 2010). The paucity of observational constraints has also limited the evaluation of emission inventories and the resulting $PM_{2.5}$ formation simulated by models. Agricultural emission inventories are often based on emission factors from animals or fertilizers under certain field conditions, which are generalized to known populations or mass applied, respectively. These conditions are highly variable due to meteorology, local livestock diet, and waste management and storage (Hristov et al., 2011). Recent studies have established that these bottom-up inventories often underestimate ammonia emissions due to difficulties in effectively scaling the low-biased measurements (Walker et al., 2012). Studies in California, in particular, show evidence of this ammonia underestimate in areas with rapidly increasing livestock populations, and they encourage improvements in ammonia emissions estimates to better predict $PM_{2.5}$ (Nowak et al., 2012; Schiferl et al., 2014). Models that underestimate the ammonia emissions will underestimate the surface $PM_{2.5}$ if sufficient acid is available, negatively affecting air quality management. However, Paulot et al. (2016) suggest that ammonium nitrate formation globally is more limited by nitric acid ($HNO_3$) than ammonia,





and that the uncertainty associated with the formation of nitric acid via $N_2O_5$ uptake has a greater impact on ammonium nitrate formation than the uncertainty associated with ammonia emissions. Regardless, as regulations in the US restrict $SO_2$ and $NO_x$, the need to understand ammonia emissions and their role in the environment is growing. This importance has been recognized as new observations of ammonia have become available over longer time periods and with more spatial coverage.

Given these new observations and their relevance to understanding inorganic $PM_{2.5}$ formation, our goal is to understand the factors that control ammonia concentrations and their variability in the atmosphere. This study uses newly available observations to investigate the variability of ammonia in the US during a five year time period (2008–2012). We first identify observed ammonia variability and investigate the ability of a chemical transport model to reproduce these observations. Then,

we attribute sources of the model ammonia concentration variability and use known relationships in an attempt to more accurately represent the variability of agricultural ammonia emissions.

## 2 GEOS-Chem Simulation

### 2.1 General Description

We use the GEOS-Chem chemical transport model (www.geos-chem.org) to simulate ammonia concentrations over the US.
The scenarios described throughout this paper are driven by GEOS-5 assimilated meteorology for 2008 to 2012 from the NASA Global Modeling and Assimilation Office. We use v9-02 of the GEOS-Chem model in a nested configuration over North America at a horizontal resolution of 0.5° × 0.667° (Wang et al., 2004; Chen et al., 2009). The chemistry and transport timesteps for these nested simulations are 20 min and 10 min, respectively. A global simulation at 2° × 2.5° horizontal resolution is used to generate the boundary conditions necessary for the nested simulations. There are 47 vertical layers in all
cases. The representation of the sulfate-nitrate-ammonium aerosol system and its relevant precursor gases in the standard version, including emissions, chemistry, and deposition schemes, generally remains as that described previously in Schiferl et al. (2014). Briefly, the coupling of gas-phase chemistry to aerosol chemistry in GEOS-Chem is described by Park et al. (2004). The gas-particle partitioning of ammonium nitrate is calculated by ISORROPIA II (Fountoukis and Nenes, 2007) as implemented by Pye et al. (2009), where the aerosols are assumed to exist on the metastable branch of the hygroscopic
hysteresis curve. Relevant modifications from v9-01-01 used in Schiferl et al. (2014) to v9-02 used here include updates to the seasonal cycle of the US Environmental Protection Agency's (EPA) National Emissions Inventory for 2005 (NEI-2005) ammonia emissions (Zhang et al., 2012) and to the algorithm controlling soil $NO_x$ emissions (Hudman et al., 2012) (described in Sect. 2.2).

### 2.2 Emissions and Emission Trends in Base Scenario

The "base scenario" referred to in this analysis incorporates modifications to the standard GEOS-Chem v9-02 simulation which have been made to the emissions in order to more accurately represent the study time period. In the base scenario, annual scale



factors applied to anthropogenic $SO_x$ and $NO_x$ emissions to capture the emissions trends over time (which end in 2010 in the standard model version) are extended uniformly spatially to 2011 and 2012 from EPA Trends data (www.epa.gov/ttn/chief/trends/). Mean anthropogenic $SO_x$, largely from power generation, and $NO_x$, largely from automobiles, emission rates over the US in summer (JJA) 2008 are 18 mg S $km^{-2}$ $s^{-1}$ and 16 mg N $km^{-2}$ $s^{-1}$, respectively. As

shown in Fig. 1, anthropogenic $SO_x$ and $NO_x$ emissions are highest in the eastern US and are often associated with rural point or dense urban sources. These emission rates decrease by 30 % and 33 %, respectively, by 2012. The majority of the magnitude of these decreases occurs in the eastern regions of the US. For 2008, anthropogenic $SO_x$ makes up 98 % of total $SO_x$ emissions, and anthropogenic $NO_x$ makes up 65 % of total $NO_x$ emissions. Other major sources of $NO_x$ with large interannual variability are soils and fertilizer use. In the entire US, these summertime emission rates vary from −23 % to +20 % of the mean from

2008 to 2012, with most of the variability occurring in the Plains and the Midwest regions. The soil and fertilizer $NO_x$ emission rates are simulated online and are controlled by a combination of nitrogen storage and meteorology (Hudman et al., 2012). In 2012, high temperatures increase soil and fertilizer $NO_x$ emissions, offsetting the decrease in anthropogenic $NO_x$ emissions (Fig. 1).

As in the standard version, our base scenario uses anthropogenic ammonia emissions from the EPA NEI-2005 inventory, which includes livestock, fertilizer, and non-agricultural sources. These emissions are for August and scaled uniformly spatially each month as determined by Zhang et al. (2012). The summer mean anthropogenic ammonia emission rate for the US is 12 mg N $km^{-2}$ $s^{-1}$. Livestock and fertilizer use compose 71 % and 15 % of this emission rate, respectively. The Plains and the Midwest exhibit higher total anthropogenic emission rates of 20 mg N $km^{-2}$ $s^{-1}$ and 19 mg N $km^{-2}$ $s^{-1}$, respectively, with larger

contributions from agriculture. The spatial distribution of these high ammonia emission regions are shown in Fig. 1. For the entire US, anthropogenic ammonia emissions make up 78 % of the total ammonia emissions in the summer. Other sources include natural emissions (16 %), biofuel (3.7 %) and biomass burning (1.8 %). Biomass burning emissions are highly variable over the study period (by a factor of two), which causes slight differences in the proportions mentioned above. In our base scenario, we use daily biomass burning emissions from the Fire INventory from NCAR (FINN) through 2012 (Wiedinmyer et

al., 2011). Given a nearly constant rate of ammonia emission and the large changes in $NO_x$ emissions mentioned above, changes in the ammonia concentrations may be driven by changes in the acid supply, which would affect gas-particle partitioning of ammonium nitrate and the overall $PM_{2.5}$ concentration.

There is no diurnal or interannual variability in the ammonia emissions in our base scenario. When we implement a diurnal

emission scaling determined by the local daily diurnal surface temperature profile, the mean surface summer ammonia concentration in the US is reduced by 12 % (1.62 ppb without versus 1.43 ppb with diurnal emission scaling). This mean value is heavily influenced by a large daily overnight decrease in concentration of 24 %, while the daytime concentration decrease is minimal, only 1 %. There is substantial uncertainty associated with any diurnal emission scaling scheme, and given its





modest impact on ammonia concentrations (particularly in the daytime) and the minimal resulting impact on seasonal mean PM$_{2.5}$ concentrations, the diurnal emission scheme is not used in this study.

## 2.3 GEOS-Chem Simulation of Ammonia in Previous Studies

A number of previous studies have evaluated the GEOS-Chem simulation of ammonia. These studies are often limited in their
comparison with ammonia observations and instead use measurements of PM$_{2.5}$ concentration and wet deposition flux, which are more commonly measured, to indirectly evaluate the model. The initial evaluation of the implementation of the gas-particle partitioning mechanism by Pye et al. (2009) reveals an underprediction of inorganic aerosol in the US, but they do not attribute this bias to problems with the ammonia emissions inventory. Zhang et al. (2012) apply an updated monthly scaling to the NEI-2005 ammonia emissions to improve the model bias in NH$_x$ (NH$_3$ + ammonium (NH$_4^+$)) based on network measurements of
wet deposition fluxes over a limited timeframe. Even with these improvements, the model remains biased high for nitric acid, ammonium, and nitrate (NO$_3^-$), which they suggest is due to excess production of nitric acid from N$_2$O$_5$ hydrolysis, though Heald et al. (2012) show that altering this uptake process does not improve the simulation of nitrate in the model. An underestimate of ammonia emissions in California is suggested by Heald et al. (2012) and Schiferl et al. (2014) using Infrared Atmospheric Sounding Interferometer (IASI) satellite measurements and aircraft measurements of ammonia, respectively.
Walker et al. (2012) also suggest that an increase in ammonia emissions in California is required to reduce the model bias compared to ammonium nitrate observations. The GEOS-Chem adjoint is used along with Tropospheric Emission Spectrometer (TES) measurements by Zhu et al. (2013) to constrain ammonia emissions over the US. They find an optimized solution which increases ammonia emissions in California and other parts of the western US and improves comparison of simulated surface concentration with observations from Ammonia Monitoring Network (AMoN) sites. Paulot et al. (2014)
also use the GEOS-Chem adjoint along with ammonium wet deposition measurements to similarly optimize ammonia emissions. These results increase ammonia emissions in California and the Midwest, consistent with underestimates described in previous studies, and decrease emissions in some regions of the Northeast and Southeast. Their optimization also suggests errors in the seasonality of emissions, particularly relating to fertilizer emissions in the Midwest.

## 3 Ammonia Observations

## 3.1 IASI Satellite Column Measurements

Recent work has shown that atmospheric ammonia concentration can be retrieved from satellite observations at thermal infrared wavelengths (Clarisse et al., 2009, 2010; Shephard et al., 2011). These retrievals provide greater spatial coverage of ammonia concentrations than current surface networks. Here we use a product from the IASI mission, which is designed to take full advantage of the hyperspectral character of the instrument (Van Damme et al., 2014a). An infrared radiance index,
calculated from a wider spectral range than previous ammonia satellite products, is converted to a total ammonia column value using look-up tables which depend on this index and the thermal contrast (temperature difference between the surface (skin)



and the air above). These look-up tables are computed using a forward radiative transfer model. The observations provide high spatial resolution (circular 12 km footprint at nadir) and up to twice-daily temporal resolution. Although there is vertical variation in the concentration sensitivity in the infrared retrieval, this information (e.g., an averaging kernel) is not available with this IASI product. However, an uncertainty estimate (retrieval error) is associated with each individual measurement. In

general, uncertainties are smaller for larger column concentration and larger thermal contrasts. These errors range from more than 100 % to less than 25 % under good conditions. This IASI product was initially used to examine both regional and global ammonia concentration variation, highlighting the influence of biomass burning events on the global scale as well as the ability to capture smaller ammonia emission features (Van Damme et al., 2014a). In Van Damme et al. (2015b), seasonal patterns and interannual variability at subcontinental scale are identified and an IASI-derived climatology of the month of maximum

columns is used to attribute major source processes. Ammonia column measurements from the retrieval scheme were also evaluated in Europe against a regional air quality model by Van Damme et al. (2014b). This comparison shows good agreement between observed and simulated ammonia column concentrations in both agricultural and remote regions, although average measured columns are higher than those simulated. When accounting for the lack of retrieval sensitivity during colder months, the observations capture the seasonality simulated in the agricultural regions. Van Damme et al. (2015a) present attempts to

validate the IASI product against in situ ammonia measurements, although these are challenging given the lack of highly spatially distributed measurements. The measured IASI columns tend to show less variability compared to surface measurements.

Our study uses data from the morning overpass (09:30 local solar time when crossing the equator) of IASI onboard the MetOp-

A satellite from 2008 to 2012. Each day is gridded by computing the mean column concentration (and other properties) weighted by relative error of the native retrievals within each GEOS-Chem horizontal grid box at the nested resolution (0.5° × 0.667°). The results of this gridding and averaging scheme are shown in Fig. 2 as the mean of all summers during the study period. We filter out retrievals with cloud cover greater than or equal to 25 % and skin temperature less than or equal to –10 °C as recommended by Van Damme et al. (2014a). Post-gridded values are filtered by removing grid boxes with greater than

75 % relative error. This filtering alters the distribution of the column concentration by removing the smallest values, as shown in Fig. 2. We calculate seasonal means as the simple arithmetic mean of all valid gridded daily values within that time period. This method weights each day with at least one valid retrieval evenly, rather than biasing the seasonal mean toward days with multiple valid retrievals in a grid box on a single day.

The gridded IASI values used in our analysis are more likely to be valid (meeting the retrieval and filtering restrictions) on warm, cloud-free days with high ammonia concentrations. The mean reported IASI concentrations are therefore biased, as low ammonia concentrations are harder to detect with confidence, and are thus often filtered out. Most valid retrievals occur during the summer, the time of highest concentration (and emissions in most areas) and better infrared retrieval conditions. As shown in Fig. 2, the range of mean (2008–2012) summer gridded and filtered concentrations is from 0.4 to $7 \times 10^{16}$ molec cm$^{-2}$. The



IASI column concentrations are highest in known agricultural regions such as the Central Valley of California, the Plains, and the Midwest. Individual spatial features are well defined and benefit from the high horizontal resolution satellite product.

Although filtered to exclude a maximum relative error, the remaining errors remain higher along the east coast and throughout the southeastern US, which has lower ammonia concentrations and lower thermal contrast. The relative error is also inversely related to the number of valid retrievals present in each grid box for a certain timeframe. These parameters are shown for comparison in Fig. 2. The hot, dry, and cloud-free conditions experienced in the western US in the summer are ideal conditions for infrared retrievals. The higher emissions and concentrations of ammonia during the summer months also yield more information and higher confidence during this time. Thus, we restrict much of our analysis and discussion to the summers of 2008–2012. The lack of an averaging kernel provided with the IASI product makes a traditional model-measurement comparison challenging. We therefore focus on the qualitative spatial and temporal constraints from IASI.

We do not use other satellite measurements of ammonia, available from TES aboard the Aura satellite, the Cross-track Infrared Sounder (CrIS) aboard the Suomi National Polar-orbiting Partnership (NPP) satellite, and the Atmospheric InfraRed Sounder (AIRS) aboard the Aqua satellite (Shephard et al., 2011; Shephard and Cady-Pereira, 2015; Warner et al., 2015). While the footprint of TES (~8 km) is smaller than that of IASI (~12 km), IASI has substantially better spatial coverage given TES's limited cross-track scanning. Thus, the measurement frequency over the same area is much higher for IASI and more useful for studying ammonia variability. The CrIS and AIRS products have only recently been developed. Further, CrIS has been active since only 2011, providing a limited timeframe for studying the variability of ammonia, and AIRS focuses on ammonia concentrations at a vertical height of 918 hPa, the location of highest instrument sensitivity, which excludes much of the western US, which is located above this height, from analysis.

### 3.2 AMoN Surface Measurements

AMoN (nadp.sws.uiuc.edu/amon/) reports integrated two-week measurements of ammonia surface concentration at fixed ground sites across the US. While 14 days is the goal measurement frequency, this can vary by up to a week in either direction. AMoN was established in 2007, and we use measurements from 2008 through 2012 in our study. The number of sites and spatial coverage of the network has increased greatly throughout this time frame (Fig. 3). Fourteen sites provide measurements for the entire study period, with 57 sites operating by 2012. Measurements are made using triplicate passive diffusion samplers, where ammonia sorbs to a phosphoric acid-coated surface. The resulting ammonium is removed via sonication and measured with flow injection analysis (Puchalski et al. 2011). The passive sampler measurements used by AMoN have a $2\sigma$ uncertainty of 6.5 % (www.radiello.com). Evaluation of these samplers against annular denuder measurements shows a consistent low bias, especially when measuring concentrations below 0.75 $\mu g\ m^{-3}$ (at 20 °C and 1 atm, which is 0.99 ppb at STP) (Puchalski et al., 2011) However we note that AMoN does not report blank corrections which could bias these measurements high (Day et al., 2012). AMoN measurements, reported in $\mu g\ m^{-3}$ are converted to ppb using local temperature and pressure from the





GEOS-5 meteorology in this study. The summer seasonal mean surface ammonia concentrations measured by AMoN ranges from 0.43 ppb in Coweeta, North Carolina to 31 ppb in Logan, Utah during our study period. When calculating seasonal mean AMoN surface ammonia concentrations, we define the date of an individual AMoN measurement as the center date of its measurement time period. AMoN measurements from 27 sites from November 2007 to June 2010 have previously been used

by Zhu et al. (2013) to evaluate the optimization of ammonia emissions used in GEOS-Chem. Their initial comparison prior to optimization showed that GEOS-Chem was generally biased low for surface ammonia concentrations throughout the year, with particularly poor performance in the spring.

### 3.3 Airborne Measurements

High resolution measurements of ammonia have recently been made in three dimensions aboard aircraft during field campaigns

throughout the US. We use data from seven campaigns, which we separate into seven regions, for a total of nine snapshots of the vertical distribution of ammonia concentration. Specific information regarding these cases, including locations, dates, instrumentation, and uncertainty, is listed in Table 1. All measurements were made with a 1 s interval, except those made during DISCOVER-AQ in California, which used a 3 s interval, and those made during ICARTT in the northeastern US, which used a 5 s interval. In all cases, the ammonia concentration measurements are averaged to 1 min time resolution. The horizontal

spatial distribution of these measurements are shown in Fig. 4a.

### 3.4 Observed Year-to-year Ammonia Variability

The observed ammonia concentration can be modulated by numerous anthropogenic and environmental factors including ammonia emissions, meteorology, and the emission of acid precursors (i.e., $SO_x$ and $NO_x$). Emissions of anthropogenic ammonia are affected by changes in agricultural activities such as livestock population and fertilizer application, as well as,

the implementation of catalytic converters in urban areas. These emissions are sensitive to meteorology that modulates volatilization from the agricultural ammonia sources, increasing with higher temperature and wind speed. Biomass burning events are highly variable and temporarily increase ammonia emissions. Our baseline simulation captures only the year-to-year variation in biomass burning emissions of ammonia; emissions from all other sectors are fixed. Meteorology affects the partitioning of ammonia into ammonium nitrate, where higher temperature and lower relative humidity favor the gas phase, as

well as the removal of ammonia from the atmosphere by changing the rates of both wet and dry deposition (Russell et al., 1983; Mozurkewich, 1993). Even in a well-mixed boundary layer, ammonia concentrations may have strong gradients caused by temperature variations with altitude that alter gas-to-particle partitioning of ammonium (Neuman et al., 2003). Figure 5 shows the year-to-year variation in key meteorological parameters across the US from 2008 to 2012 from the GEOS-5 assimilated meteorological product. Emissions of $SO_x$ and $NO_x$ also affect the ammonia concentration by regulating the amount

of acid available to convert ammonia into ammonium sulfate and ammonium nitrate particles. Figure 1 shows that the anthropogenic component of these emissions decreases substantially in the US during our study period. Meteorology can also

affect the rate of soil and fertilizer $NO_x$ emissions by changing the storage and volatilization processes (simulated changes also shown in Fig. 5).

Using IASI column concentration and AMoN surface concentration measurements, we show in Fig. 3 that observed ammonia concentrations vary significantly from year-to-year over the US. The mean IASI column concentration observed over the US in the summers of 2008 through 2012 is $0.95 \times 10^{16}$ molec cm$^{-2}$, which ranges from a low of $0.90 \times 10^{16}$ molec cm$^{-2}$ in 2010 to a high of $1.1 \times 10^{16}$ molec cm$^{-2}$ in 2012 (indicating that the mean ammonia column concentrations over the US range from $-5.3$ % to $+16$ % of the mean during these five years). At the surface, the mean AMoN observed ammonia concentration in the summer from all sites with records from 2008 to 2012 is 3.4 ppb, ranging from 3.0 ppb in 2009 to 4.3 ppb in 2012 (or between $-11$ % and $+25$ % of the mean). The IASI and AMoN observations differ on the year with the lowest mean summer concentration (2010 for IASI and 2009 for AMoN); this difference is likely due to a lack of AMoN sites distributed throughout areas which have low IASI column concentrations in 2010. The regions of high agricultural production, including California and the Plains, exhibit higher year-to-year variability in the magnitude of IASI column concentrations. For example, in the Plains region, maximum summer IASI values are 23 % higher in 2012 than the mean of the five study years. This is also the case for surface concentrations at several AMoN sites in the Midwest and the West.

In what follows, we will use the GEOS-Chem model to examine the source of the observed year-to-year variation in ammonia concentrations.

## 4 Base Scenario Simulation of Ammonia Measurements

Throughout this section, we use the GEOS-Chem model to investigate how well the model captures the observed magnitude and variability in ammonia concentrations. We sample the model to simulate the ammonia concentrations observed in both temporal and spatial dimensions.

### 4.1 Column Comparison

To evaluate the ammonia concentration throughout the column, the simulated column concentrations are recorded at the local 09:00–10:00 overpass time, and this one-hour mean is compared to the IASI retrievals at 09:30 local time. It is not straightforward to compare this value in an unbiased way with the IASI measurements since the vertical sensitivity of the instrument may not be consistent with the model. For this reason, this value cannot be quantitatively compared to the IASI retrieved column with confidence, however we qualitatively compare trends and spatial features here. When sampling is applied, only simulated days with valid IASI retrievals (at least one per grid box) are included. Seasonal means are calculated as the mean of all days (no sampling) or of only days with valid IASI retrievals (with sampling). The simulated ammonia column concentrations are generally well correlated with the IASI observations (Fig. 6) over the summer, particularly in the





Plains and the Midwest (correlation (R) = 0.6–0.8). Sampled simulated column concentrations shown in Fig. 3 have a summer mean of $0.64 \times 10^{16}$ molec cm$^{-2}$, ranging from $0.52 \times 10^{16}$ molec cm$^{-2}$ in 2009 to $0.80 \times 10^{16}$ molec cm$^{-2}$ in 2012 (or between −19 % and +25 % of the mean).  We find considerable year-to-year variation in the simulated ammonia concentration, even with fixed ammonia emissions.

Sampling the model to match IASI observations, as shown in Fig. 3, increases the concentrations in regions with more invalid IASI days according to the filtering process described in Sect. 3.1. Valid days tend to have higher concentrations as they meet the filter requirements due to more favorable retrieval conditions, which include a higher retrieved ammonia signal. Cloudy days, being cooler and having greater probability of rain, also tend to have lower ammonia concentrations, and these cannot
be retrieved. In the southeastern US, sampling increases the regional summer mean simulated ammonia column concentration significantly, by 26 % (2011) to 58 % (2012). Even after accounting for this sampling bias, the simulated column concentrations are consistently lower than those observed by IASI, which is consistent with the findings of Van Damme et al. (2014b) over Europe. This underestimate is because the filter requirement restricting high relative error inherently favors larger observed columns. Consequently, there is lower year-to-year variability in the mean summer IASI column concentrations (21
% of the mean between highest and lowest years) than those simulated by the model (44 %).

The distribution of ammonia throughout the column is also relevant to assessing the ability of the model to represent the ammonia column concentration observed by IASI, as the retrieval has varying sensitivity at different vertical levels. In Fig. 4b we use measurements of ammonia from several aircraft campaigns throughout the US to evaluate the simulated ammonia
vertical profile. We show the median, rather than the mean, to account for the inherent inability of the model to reproduce highly-concentrated plumes occasionally observed by the aircraft. To compare the observations with the simulation during campaigns that take place in our study period (extended to February 2013), we sample the model directly in time and space for each flight of the campaign. For campaigns outside of this time period, we sample directly in space for each flight but approximate the time component by using the five-year mean (2008–2012) of each two-month campaign window. As shown
in Fig. 4b, the observed median ammonia vertical profile is highly variable in magnitude and shape between different regions. In high ammonia emission regions, the observed ammonia concentration increases greatly toward the surface, and the median ammonia vertical profile is less variable between different campaigns in the same region (e.g., Central Valley in 2010 and 2013, Colorado in 2014 and 2015) than between different regions. As with the observations, the model performance varies greatly between regions. Over areas such as the Central Valley, previously examined by Schiferl et al. (2014), the model
underestimates ammonia throughout the vertical profile, especially near the surface. The model also performs more poorly in the spring according to measurements in Colorado and the southern Plains in 2015, but limited sampling across seasons makes it difficult to be conclusive. Other regions, like southern California, eastern Texas, Colorado in summer 2014, and the southeastern US have a much smaller bias. The slight high bias in the model at the surface in the northeastern and southeastern US regions is consistent with previous evaluation of NEI-2005 in GEOS-Chem against AMoN measurements (Paulot et al.,



2014). Local conditions clearly influence the model simulation of the observed concentrations. Overall, the model shows less variability than the observations, but the model profile shape is generally consistent with the observed shape outside of large source regions. This suggests that, outside of source regions, model biases in the shape of the vertical profile are unlikely to bias comparisons with satellite column observations.

## 5    4.2 Surface Comparison

Summer seasonal mean simulated surface concentrations are compared with the seasonal mean AMoN surface concentration observations in Fig. 3. For a more direct comparison of individual observations, we match the hours of the AMoN sampling period with the corresponding hourly values from the simulation, and the mean of these hours is used for comparison. We also apply a spatial interpolation scheme to this comparison, where the four nearest grid box values are averaged based on the
distance between their center and the observation site location. This adjusts the simulated concentration to account for the influence of nearby grid boxes at sites near grid box edges and in regions which exhibit strong horizontal gradients. The mean summer simulated surface concentration at AMoN sites with measurements from 2008 to 2012 (11 sites) is 2.5 ppb, which varies from a low of 2.1 ppb in 2008 (–16 % of the mean) to a high of 2.8 ppb in 2012 (+13 % of the mean) over the study time period. This mean simulated concentration is lower than that observed (2.8 ppb versus 3.4 ppb). The range of simulated
surface concentrations between high and low years is also half of the range observed (0.73 ppb versus 1.3 ppb). These ranges are shown for comparison in Fig. 7 along with the range in surface ammonia concentrations over the entire US. The range in summertime mean ammonia concentrations across the US is smaller, and the mean is lower (by more than 25 %) than when sampled to the AMoN sites. This suggests a sampling bias for the AMoN network as a whole, as many AMoN sites are located near high ammonia source regions. The near-source location of many of these AMoN sites provides an additional challenge
for the regional-scale resolution model simulation used here and is likely responsible for some of the model underestimate.

By limiting the above analysis to only summers 2011 and 2012, the number of sites with measurements in both years increases to 48. The mean bias in this case is more modest (–0.02 ppb), with 2011 biased slightly high and 2012 biased slightly low. There is a consistent high bias at many of the eastern US sites, which is offset by a low bias in the West in 2012, likely due to
local biomass burning which is not adequately captured in the model. However, even for this limited time period, the model fails to reproduce the observed year-to-year variation (observed 0.80 ppb increase in the summertime mean from 2011 to 2012, with a simulated increase of only 0.11 ppb). This difference is dominated by high measurements in 2012 in the West, but the observed increase from 2011 to 2012 in the Midwest is also underestimated.

Figure 8 shows a detailed comparison of observed and base scenario simulated surface ammonia concentrations at three AMoN sites with records from 2008 to 2012; these are selected as representative regional sites and demonstrate the varying degree of model skill. Simulated concentrations at all three sites reproduce the observed seasonal cycle, with highest concentrations in the summer and lowest in the winter. The Indianapolis, Indiana site represents typical Midwestern sites, with nearby urban



SO$_x$ and NO$_x$ emission sources surrounded by rural ammonia sources. This site is located in central Indianapolis, and the corresponding model grid box is made up of about 30 % city and 60 % rural land. The overall comparison at Indianapolis is good throughout the study period, with an R of 0.56 and normalized mean bias (NMB) of –0.14 (mean bias of –0.41 ppb). There is a noticeable increasing trend in the observed ammonia concentrations from 2008 to 2012; the model captures much of this upward trend.

The Horicon Marsh, Wisconsin site represents rural regions where ammonia emissions are primarily from agricultural sources. This site is located in a grid box which is nearly 90 % farm land (the remaining 10 % is made up of small towns and wetlands). This uniformity should be easier for the model to represent. The comparison between observed and simulated ammonia concentration is generally very good when considering the entire time period (R = 0.65, NMB = 0.06, mean bias = +0.19 ppb). However, this comparison is somewhat worse in the summer (R = 0.44) as the model does not properly simulate the timing or magnitude of the peak concentrations.

Finally, the Fort Collins, Colorado site represents one of several sites in the western US which present a challenge to simulate due to large horizontal concentration gradients over areas with highly varying topography. This is an area of high livestock ammonia emissions to the east bounded on the west by the Front Range of the Rocky Mountains. Ammonia is advected from feedlots to the east and observed high concentrations result. The site is located on the eastern side of a grid box which is made up of 75 % mountains and forest toward the west. There is considerable elevation increase as well from east to west. As a result, simulated concentrations in this grid box take on the characteristics of the mountain region, rather than agricultural plain. There is a large low bias at the Fort Collins site of –4.7 ppb (R = 0.50, NMB = –0.77) for the entire time period. If we compare the observations with simulated values of the next grid box east in the agricultural region (without weighting neighboring grid boxes) the bias drops significantly (about 35 %), so that only –3.0 ppb bias in all months remains. Even with this adjustment to account for site location, the model performance here is among the poorest.

## 4.3 Integrated Comparison: Colorado, Summer 2012

The variation of both the observed column and surface ammonia concentrations in the western US is influenced by biomass burning events in the summer of 2012. The wildfire activity in the Colorado Front Range during this time (May–September 2012) provides an opportunity to synthesize the different ammonia concentration information discussed above as this is an area which is also known for high agricultural ammonia emissions.

IASI measurements during days without fire emission influence (determined by visual inspection of Moderate Resolution Imaging Spectroradiometer (MODIS) imagery, with at least 75 % domain retrieval coverage) show a peak mean column concentration of $2.1 \times 10^{16}$ molec cm$^{-2}$ just to the east of Fort Collins (FC) (Fig. 9), corresponding to the location of feedlots. The mountains to the west of FC, along with ridges to the north and south, cause the agriculturally emitted ammonia to circulate





throughout the Front Range, with only limited transport westward (Wilczak and Glendening, 1988). Column concentrations remain elevated to the south and east throughout the plain of eastern Colorado, while concentrations in western areas of the domain at high elevations are quite low. Aircraft measurements in Colorado during the FRAPPE (summer 2014) and SONGNEX (spring 2015) campaigns confirm this distribution of ammonia in the region (Fig. 4a). Figure 9 shows that IASI

column concentrations are considerably higher on days with wildfire activity. The largest increase takes place over the Front Range near FC and to the east due to fires located in the Colorado mountains during late June–early July, when the mean IASI column over the region more than doubles. In August, column concentrations are enhanced in the north and west of the domain due to the transport of wildfire plumes into the region from fires in other areas of the northwestern US. Thus, we see in Fig. 9 that the average ammonia concentrations observed by IASI during the season are elevated throughout the region due to fire

emissions. These wildfire emissions are present in addition to the persistent agricultural ammonia sources throughout the time period, as the feedlot grid box east of FC has the highest column concentration even on wildfire-influenced days ($3.4 \times 10^{16}$ molec cm$^{-2}$, increase of 62 %). However, the IASI retrieval is more sensitive to ammonia lofted vertically, as is the case in biomass burning outflow. The GEOS-Chem simulated ammonia column concentrations in this domain do not capture the peaks observed by IASI throughout the time period. This suggests that the model inventory underestimates the fire emissions of

ammonia or their injection height; these biases are likely exacerbated by the IASI vertical sensitivity.

AMoN surface concentrations at the FC site, also in Fig. 9, follow the peaks in concentration observed by IASI in both June and August and show a similar relative increase (factor of ~2 in late June), while surface concentrations at the Longs Peak (LP) AMoN site show no evidence of fire influence as the site is isolated from the Front Range source region. It is difficult to

quantify the contribution of the wildfire ammonia source from these observations because the fire events also correspond with the highest surface temperatures of the year, thereby affecting ammonia volatilization and partitioning chemistry. Additional observations of ammonia concentrations in fire plumes could help improve emissions estimates and clarify the importance of this source (e.g., Whitburn et al., 2015).

**4.4 Updated Inventory Comparison**

A more recent anthropogenic emission inventory, NEI-2011, is available over the US for 2011 (available from www3.epa.gov/ttnchie1/net/2011inventory.html, adapted for GEOS-Chem by Travis et al. (2016)). This inventory includes changes in both the magnitude and timing of anthropogenic ammonia, $SO_x$, and $NO_x$ when compared to NEI-2005. Averaged over the summers during the study period of 2008 to 2012, anthropogenic ammonia emissions are 26 % higher, anthropogenic $SO_x$ emissions are 13 % higher, and anthropogenic $NO_x$ emissions are 11 % lower in NEI-2011 compared to in NEI-2005 as

applied to GEOS-Chem over the US. Variable spatial seasonality for ammonia emissions has been included in NEI-2011 such that known emissions events like springtime fertilizer application in the Midwest are now accounted for.





We repeat our GEOS-Chem simulations with NEI-2011 for 2008 and 2012 and compare the simulated surface concentrations with the observed AMoN surface concentration in these two years. Generally, the summer high concentration bias at the eastern US sites is reduced using the updated inventory. The simulation improves at a few of the western sites as well, but many biases remain, due especially to the inability to reproduce peak concentration values. Strong gradients in local sources and geography

still likely play a large role at many of these sites. At Midwestern sites, the new seasonality often better represents the springtime and summer peak concentration, but the comparison during the transition to late summer and fall is degraded. For Horicon Marsh, Wisconsin, the summer R in 2008 and 2012 between observed and simulated surface concentration decreases from 0.63 to 0.48 when using NEI-2011 rather than NEI-2005. While NEI-2011 may better represent the magnitude and timing of emissions in some locations, it is also a year-specific inventory and does not provide a better constraint than NEI-2005 on

the year-to-year variations in ammonia emissions that is the main focus of this study.

### 4.5 Summary of Base Scenario to Observation Comparisons

From the comparisons described here, we conclude that the model generally captures the vertical, temporal, and regional variability of ammonia but underestimates the summertime ammonia concentration observed in both the column and at the surface, particularly near source regions (including both agricultural and fire emissions). The year-to-year variability in the

model at the surface is lower than the variability observed, but the trends and variability captured by the simulation are significant considering that ammonia emissions in the model are fixed. We next explore the processes in the model which contribute to this variability.

## 5 Attributing Sources of Ammonia Variability

### 5.1 SO$_x$ and NO$_x$ Emissions Reductions

In order to identify the drivers of year-to-year variation in simulated ammonia concentrations, we run sensitivity studies which isolate individual factors affecting the ammonia concentrations. The first sensitivity simulation holds anthropogenic SO$_x$ and NO$_x$ emissions constant at 2008 levels for 2009 to 2012 in order to gauge the effects of these emissions reductions on the ammonia concentration in the base scenario.

Figure 10 shows that SO$_x$ and NO$_x$ reductions over the US act to significantly increase the ammonia column concentration over time. Much of this increase takes place over the eastern US, where anthropogenic SO$_x$ and NO$_x$ emissions are highest (Fig. 1), and therefore where absolute reductions in SO$_x$ and NO$_x$ are largest. Decreases in the sulfate and total nitrate (TNO$_3$ = HNO$_3$ + NO$_3^-$) availability caused by the SO$_x$ and NO$_x$ emission reductions, respectively, require less ammonium to neutralize particle phase acids, leaving more ammonia in the gas phase. For the US summer mean, the simulated ammonia

surface concentrations increase by 8.8 % from 2008 to 2012 due to the anthropogenic emissions changes, compared to the 29 % decrease in total SO$_x$ emissions and the 17 % decrease in total NO$_x$ emissions. We attribute 32 % (0.17 ppb) of the range of





summer surface ammonia concentration simulated by the base scenario to anthropogenic $SO_x$ and $NO_x$ emissions reductions. In the column, 26 % ($0.07 \times 10^{16}$ molec cm$^{-2}$) of the range is due to these reductions.

## 5.2 Meteorology Variability

The second sensitivity simulation tests the effects that meteorological variability has on the simulated ammonia concentration. In this simulation, we hold the GEOS-5 assimilated meteorology constant at year 2008 conditions for all years of our simulation (2008–2012). Meteorology can alter the distribution and phase of ammonia via changes in transport, deposition, oxidation, and gas-particle partitioning. Soil and fertilizer $NO_x$ emissions are also effectively held constant in this simulation given that their variability is largely controlled by meteorology. While meteorology may indirectly affect biomass burning emissions, such as by leading to more fires during a dry and hot year, we do not account for this here, as these emissions are allowed to vary in all cases. Comparison with both 10- and 35-year mean Modern-era Retrospective Analysis for Research and Applications (MERRA) meteorology from the NASA GMAO (Rienecker et al., 2011) shows that 2008 is a typical meteorological year in the US. Thus, anomalies from 2008 in 2009–2012 can be seen as realistic deviations from an average condition.

Figure 10 shows that the effects of meteorology on the ammonia concentration are highly variable both spatially and temporally. The spatial variability is generally greater at the surface (not shown) than in the column. Variations in simulated ammonia concentration can be connected with the meteorological features shown in Fig. 5. For example, the summer of 2010 in the southeastern US is a high-precipitation year which contributed to lower ammonia concentration throughout the column due to increased wet removal. Higher relative humidity also likely contributes to this decrease by favoring the particle phase of the ammonium nitrate equilibrium. Another example is the high-temperature, low-humidity and low-precipitation summer of 2012 in the Plains and the Midwest, which favors the gas phase of the ammonium nitrate equilibrium and generally higher concentrations (due to reduced removal). However these same high temperatures in 2012 lead to higher emissions of soil and fertilizer $NO_x$, which modestly counteract this effect at the surface by encouraging more ammonia to partition to the particle phase to neutralize this supply of acid (Fig. 5). Lower planetary boundary layer (PBL) heights, such as in the upper Midwest in summer 2011, can trap ammonia near the surface. More ammonia nearer the surface could increase the dry deposition flux as this is the primary direct removal method for gaseous ammonia, slightly offsetting the increased concentration due to trapping and decreasing the concentration throughout the column. We attribute 64 % (0.34 ppb) of the range of summer surface ammonia concentration simulated by the base scenario to meteorology. In the column, 67 % ($0.18 \times 10^{16}$ molec cm$^{-2}$) of the range is due to these variations. Meteorology clearly dominates the year-to-year variability in simulated ammonia concentration.



### 5.3 Missing Simulated Ammonia Variability

The simulated ammonia concentrations do show significant year-to-year variability despite constant ammonia emissions, but this variability is generally lower than that observed by IASI and AMoN at individual locations (Figs. 3 and 7). However, maximum observed column concentrations in the western US in 2012 are likely from smoke enhancements at the vertical
levels at which IASI is more sensitive; the model cannot reproduce this column variability without properly weighting the different vertical levels sensitive to these concentrations. There are also not enough AMoN sites over the entire time period to robustly indicate either regional variations in surface ammonia concentration or whether a particular site is impacted by local emission changes. The range of simulated mean ammonia concentrations is 0.53 ppb less than the range observed at the available sites over the summers of 2008 to 2012 (Fig. 7). Most of this missing range is from sites in the West and the Midwest,
where agricultural ammonia emissions are higher. The observed range is likely influenced by high biomass burning emissions in the West and high temperature effects on partitioning in the Plains and the Midwest, which are greater than in the model. In addition, the base scenario does not account for variations in year-to-year changes in agricultural ammonia emissions, so we next assess how much influence these variations may have on the ammonia concentration.

## 6 Implementing Agriculture Ammonia Emissions Variability

### 6.1 Activity Scaling

The base scenario anthropogenic ammonia emissions are constant for all years of study. This is not realistic due to year-to-year changes in agricultural activity and the meteorological dependence of emissions (Sect. 6.2). We define agricultural activity as livestock population and fertilizer application. Using data from the US Department of Agriculture National Agricultural Statistics Service (USDA NASS) (www.nass.usda.gov), we compute annual scale factors for agricultural activity based on the
changes in these sources (for description of methods see Sect. A1 in Appendix A).

As shown in Fig. 11, this scaling results in large increases in livestock ammonia emissions compared to the base scenario in Iowa (13 % by 2012), although this is relatively constant during the study period (only 2.3 % increase between 2008 and 2012). The more dramatic change occurs over Texas and Oklahoma where livestock populations, largely beef cattle, decrease by 18
% between 2008 and 2012 with a net loss of 20 % compared to the base year by 2012. This large decrease in beef cattle population is due to extended extreme drought which reduce cattle food supply and force higher cull rates (Peel, 2012).

The changes in ammonia emissions due to fertilizer application variations are smaller than those for livestock population (Fig. 11). There is a noticeable decreasing trend in fertilizer application in the Texas and Oklahoma region due to a decrease in crop
planting during the drought mentioned above (18 % loss between 2008 and 2012), and an increase in the northern Plains of 20





% compared to the base year by 2012. Our approach likely underestimates the year-to-year variation in fertilizer ammonia emissions in the Midwest (see Sect. A1 in Appendix A for details).

Although some locations experience large changes in total anthropogenic ammonia emissions due to activity variations (e.g. – 13 % in Texas and Oklahoma), the US mean change is only about –2.5 %. This is consistent with the EPA Trends data, which suggests a 3.0 % decline in ammonia emissions between 2008 and 2012. Our changes present a spatial distribution of these shifts, however, rather than one national trend value.

### 6.2 Volatilization Scaling

The anthropogenic ammonia emissions in the base scenario also do not account for changes in the transfer of ammonia from the surface to the atmosphere due to temperature and wind speed variability (referred to together here as changes in volatilization). Higher temperatures (increased volatility) and greater wind speeds (increased transport) lead to higher ammonia emissions. We compute monthly scale factors which account for the effects of temperature and wind speed on both livestock and fertilizer emissions. This generally follows the methods used by Paulot et al. (2014) for the Magnitude and Seasonality of Agricultural Emissions model for $NH_3$ (MASAGE_NH3) and is described in Sect. A2 of Appendix A.

The changes in ammonia emissions computed from volatilization scaling are overall smaller, but they are more spatially variable compared to those due to agricultural activity (Fig. 11). The scenario with volatilization scaling increases US mean summertime ammonia emissions by 0.1 % in 2012 and decreases emissions by 3.2 % in 2009 compared to the base scenario. Together, activity and volatilization scaling add 2.8 % variability compared to the mean of the base scenario over the US. This variability is largest over the Midwest (6.4 %) and the Texas and Oklahoma (14 %) regions.

### 6.3 Resulting Changes to Ammonia Concentration

We simulate the ammonia concentrations for two cases as described above: 1) with added activity (livestock + fertilizer) variability of ammonia emissions and 2) with both activity and volatilization variability of ammonia emissions. The results from these simulations are shown for column concentrations in Fig. 12. Changes for volatilization alone are calculated as the difference between the two scenarios (not shown). Nearly all changes to the ammonia concentrations follow directly from changes in the ammonia emissions since summer meteorology generally favors the gas phase of the ammonium nitrate equilibrium. Thus, changes for both simulations are of similar magnitude, with more spatial and temporal variability caused by incorporating volatilization variability into the emissions. Activity emission variability decreases the mean US summer column by only $0.01 \times 10^{16}$ molec cm$^{-2}$ (2 %) throughout 2008 to 2012 compared to the base scenario, and adding volatilization variability has no further effect on this mean. Activity and volatilization variability oppose one another, leading to a net decrease of only $0.01 \times 10^{16}$ molec cm$^{-2}$ (4 % of the base scenario range). Summertime R between daily IASI observations and the simulated column concentrations in 2011 and 2012 increases by up to 0.1 in the Midwest, but decreases by a similar





magnitude in Texas and Oklahoma (compared with base scenario magnitude R in Fig. 6). At the surface, activity and volatilization emission variability decreases the mean US summer concentration by similarly small proportions (1−2 %) and has a limited effect on the range of values between minimum and maximum year surface concentrations for this domain (Fig. 7). The largest changes in surface ammonia concentration take place where the largest emission changes occur. In Texas and

Oklahoma, ammonia concentration decreases by 0.5 ppb or 17 % of the base scenario for summer 2012, the year with the largest changes.

The "best" scenario (including both activity and volatilization emission variability) also does not greatly improve the simulation bias or range compared to AMoN observations. For the sites with observations from 2008 to 2012, the scenario

with activity and volatilization agricultural ammonia variability further degrades the simulation in summertime, increasing the bias from −0.93 ppb to −1.02 ppb (Fig. 7). For the 2011–2012 timeframe when more sites are available, the magnitude of the mean summer bias increases from −0.02 ppb to −0.07 ppb. This is likely skewed toward the numerous low-concentration sites in the eastern US which start observing in 2011. However, variations in the ammonia emissions do moderately improve the ability of the model to capture year-to-year variations in surface ammonia concentrations measured at some AMoN sites, with

increases in R during the entire study period of up to about 0.07 (mean increase of 0.01). At Horizon Marsh, Wisconsin, the R between observation and model improves from 0.65 to 0.67 in all seasons, but from 0.44 to 0.54 in summer only.

We find that year-to-year variations in regional ammonia emissions play a modest role in controlling observed variations in summertime ammonia concentrations. Our simulation including this variation remains biased compared to observations

throughout many regions of the US. There are several factors that may contribute to the remaining simulation bias of ammonia concentration magnitude and variability compared to the observations. Much higher spatial resolution may be required to adequately capture ammonia concentrations in areas with high horizontal concentration gradients (see Fig. 4a); however given the sparse coverage of the AMoN network it is challenging to assess the role that site placement plays in biasing our comparisons. Additionally, better observational constraints, such as satellite products with vertical sensitivity information

could help identify the source of bias in the model.

## 7 Impacts of Ammonia Variability on Surface PM$_{2.5}$ and Nitrogen Deposition

Ammonia neutralizes acids in the atmosphere to produce PM$_{2.5}$ under appropriately cool and humid meteorological conditions. Changes in ammonia emissions, acid-precursor emissions, climate, and meteorology may all influence the surface PM$_{2.5}$ concentration. The potential for further formation of PM$_{2.5}$ (defined here as the sum of ammonium, sulfate, and nitrate) can be

described by the gas ratio (GR) (Ansari and Pandis, 1998), as defined by Eq. (1):

$$GR = \frac{[NH_x] - 2[SO_4^{2-}]}{TNO_3} . \tag{1}$$



The seasonal mean GR over all five years (2008–2012) as simulated by our GEOS-Chem base scenario is shown over the US in Fig. 13. A GR > 1 indicates little potential for further ammonium nitrate formation given additional ammonia emissions, while 0 < GR < 1 indicates that this potential does exist, under the appropriate meteorological conditions. None of the simulated seasonal mean GR values are below zero, which would indicate incomplete neutralization of sulfate.

In the summer, we find that the surface $PM_{2.5}$ concentration is weakly sensitive to ammonia emission changes described in Sect. 6 (–0.6 % in summer 2012 compared to the base simulation) (Fig. 14). The gas phase of the ammonium nitrate equilibrium is favored under summer meteorological conditions, and the GR values in Fig. 13 show that ammonium nitrate formation potential exists only in the Inter-mountain West. Thus, nearly all change (89 %) in the $NH_x$ concentration from changing ammonia emissions remains in the gas phase. There is essentially no change in ammonium sulfate as all sulfate in ammonia emissions regions has already been neutralized (GR > 0). Rather, Fig. 14 shows that changes in the surface $PM_{2.5}$ are driven by anthropogenic $SO_x$ and $NO_x$ emission reductions (34 % $PM_{2.5}$ reduction from 2008 to 2012) and meteorology.

Although changes in ammonia emissions are much smaller in the winter, both the meteorological and chemical conditions promote a higher potential for $PM_{2.5}$ formation in certain regions. Figure 13 shows that winter is chemically unique such that there is potential for ammonium nitrate to form throughout the eastern US should ammonia emissions increase. Averaged over the entire US, 78 % of the change in the $NH_x$ concentration from changing ammonia emissions remains in the gas phase during the winter in our final simulation which includes ammonia emissions variability. This value remains fairly high since most of the change in ammonia emissions occur in the area of GR > 1 (Plains) during the winter (Fig. 13). However, as $SO_x$ and $NO_x$ emissions decrease throughout the study period, this area where GR > 1 expands, reducing ammonium nitrate formation potential (not shown). Given the potential for ammonium nitrate formation, it may be more important to understand the variability of ammonia emissions during the winter (coldest temperatures, lowest ammonia emissions) to accurately simulate $PM_{2.5}$. Unfortunately, this is the time period when infrared satellite data exhibit the lowest sensitivity.

The spring (MAM) and fall (SON) seasons (which are colder, but with more moderate ammonia emissions) represent transition periods when ammonium nitrate may form under certain conditions (e.g., Chow et al., 1994). Although the distribution of GR is generally consistent with summer during both seasons as a whole, Fig. 13 shows that this potential ammonium nitrate response to changing ammonia emissions may exist just south of the Great Lakes. Examination of GR during individual months shows that the transition to GR > 1 in the eastern US occurs between March and April, and the reverse happens between October and November. This further narrows the range of time when ammonium nitrate formation may respond to ammonia emissions changes.

The reduction of $NO_x$ emissions dominates changes in the total simulated nitrogen (N, sum of ammonia, ammonium, nitric acid and nitrate) over our study period and results in a total summertime N deposition decrease of 12 % from 2008 to 2012. In





the base scenario, this decrease is partially offset by meteorological-driven factors which increase $NO_x$ emissions in 2012. The $SO_x$ and $NO_x$ emission reductions create no net effect on total $NH_x$ deposition, but there is a shift away from the particle phase flux (ammonium) toward the deposition of the gas phase (ammonia). As the simulated lifetime to total deposition of ammonia is shorter than that of ammonium (2.6 days versus 7.5 days over the US in summer 2008), this shift in phase preference

decreases the overall lifetime of $NH_x$. The shortening of the $NH_x$ lifetime to deposition means that reduced N from agricultural sources will deposit closer to the source, perhaps reducing required fertilizer inputs, but also putting sensitive ecosystems located close to source regions at risk.

Meteorology greatly influences the variability in the magnitude of $NH_x$ deposition. Simulated summertime $NH_x$ deposition

flux is dominated by gas phase ammonia, rather than particle phase ammonium. The summertime ammonium deposition which does occur is largely removed via wet processes, which is more sensitive to meteorology changes than to ammonia emissions changes. In the winter, ammonium deposition dominates the total $NH_x$ deposition flux; however, changes during this season may not be representative of the entire year as only 11 % of US agricultural ammonia emissions in our base scenario occur during the winter, compared to 36 % in summer. Together, these results indicate that wet ammonium deposition may not

always be a good proxy for ammonia emission changes. This is especially true in dry locations or during particularly dry summers, which in turn also have higher ammonia emissions.

## 8 Conclusions

We used a combination of surface, column, and aircraft ammonia concentration measurements along with a chemical transport

model to assess simulated ammonia concentrations and analyze the variability of ammonia over the US from 2008 to 2012. The model often underestimates the observed ammonia concentrations at the surface and those measured by aircraft throughout the column, however these observations are most often located near large source regions. The model performs well in areas of lower observed concentrations, such as in the eastern US. The observed seasonality at the surface is well captured by the model, outside of the timing of springtime fertilizer application. However, concentration gradients are more difficult to

represent, both horizontally and vertically, as the model is not able to simulate plumes of observed high concentrations.
The simulated concentrations are generally less variable than the observed year-to-year concentrations, but this variability is larger than previously expected given constant ammonia emissions in the model. The variability in simulated ammonia concentrations is largely driven by changes in meteorology, and including year-to-year variation in ammonia emissions from agricultural sources has minimal impact on this variability. This suggests that year-specific agricultural emissions are not

critical to the simulation of summertime ammonia and $PM_{2.5}$ in regions which are not experiencing dramatic changes in agricultural activity. Summertime $PM_{2.5}$ formation is relatively insensitive to ammonia emissions changes, but the impacts of



ammonia emission changes may be more important in cool conditions such as wintertime livestock emissions and spring crop planting.

The large role that meteorology plays in controlling atmospheric ammonia concentrations (coupled to the dynamic gas-particle partitioning) suggests that it can be challenging to use a global model to test simulated ammonia concentrations, understand how these concentrations correlate spatially to emissions sources, and assess whether emissions controls have led to expected trends in ammonia concentration. Indeed, changes in observed atmospheric ammonia concentrations may often be a poor proxy for changes in ammonia emissions. These challenges support the need for better observing systems for ammonia to test regional simulations, including dense satellite observations with a quantitative description of the instrument sensitivity and more monitoring sites distributed across source and background regions.

## Appendix A: Description of Ammonia Emission Scaling Methods

### A1 Activity Scaling

Scaling of agricultural activity refers to the influence changing livestock population and fertilizer application has on ammonia emissions. For livestock population, we use data from the USDA NASS for cattle, goats, chickens, hogs, and sheep. The portion of beef cattle versus dairy cattle is determined by the ratio of beef cows to dairy cows. The census population of each species per county is gridded to the nested simulation grid box resolution for 2002, 2007, and 2012 to obtain the animal density in each grid box. We weight each species density by its relative emission factor (emission per head) to calculate the emissions value per grid box (Pinder et al., 2004; Faulkner and Shaw, 2008; Velthof et al., 2012; Paulot et al., 2014). Linear interpolation of effective emission is applied between census dates, and the emission for each year is scaled against the base year of 2005, as that corresponds to the NEI-2005 used in the base scenario, to achieve an annual scale factor for livestock population. These scale factors are applied to the livestock portion of the anthropogenic ammonia emissions.

We use data for county-wide fertilizer expense (gridded to nested resolution grid boxes) and the national fertilizer price index from the USDA NASS to develop annual scaling factors for fertilizer application. Fertilizer expense census data is available for 2002, 2007, and 2012. Each of these years is matched with the fertilizer price index (price per mass) for that year to calculate the total fertilizer mass purchased in each grid box. We assume all fertilizer purchased is applied to a field or a similar fraction of fertilizer purchases are left unused in each year. Fertilizer mass for each of these years is interpolated linearly and then scaled in comparison to 2005 values as with livestock population above. These scale factors are then applied to the fertilizer portion of the anthropogenic ammonia inventory.

One weakness in scaling the base NEI-2005 is that the emissions for that inventory are specified for August, when fertilizer application is low, and thus there is limited fertilizer magnitude to scale in the Midwest. Fertilizer emissions from NEI-2005



in the Midwest make up about 5 % of total anthropogenic emissions in that region at all times. In NEI-2011, however, fertilizer emissions make up about 10 % of total anthropogenic emissions in August, and this increases to about 30 % for summer and about 60 % for spring. Any fertilizer activity scale factor applied to NEI-2005 in the spring and summer will have a much smaller effect on the magnitude of the fertilizer ammonia emissions than if applied to NEI-2011, and thus our scaling on

fertilizer emissions is likely to be underestimated. Resulting emission magnitude changes are shown in Fig. 11.

**A2 Volatilization Scaling**

Scaling of due to volatilization refers to the effects temperature and wind speed have on ammonia emissions from both livestock and fertilizer sources. We develop monthly scale factors (individually for all five years) to approximate these effects. This procedure generally follows the methods used by Paulot et al. (2014) for MASAGE_NH3. Emissions magnitudes are not

needed, since we scale all variability to the 2005 base year. Therefore, we weight each emission source by the relative importance of temperature and wind speed. Fertilizer ammonia emission (E) is similarly dependent on temperature and wind speed everywhere, and is represented by Eq. (A1) (Søgaard et al., 2002):

$$E = 1.02^T \times 1.04^w \ , \tag{A1}$$

where the 2 m temperature (T) and 10 m wind speed (w) values used in the calculation are from the GEOS-5 meteorology used

in the simulation. Livestock manure emissions vary differently depending on location of the manure: application, housing, or storage. The application portion varies as fertilizer above in Eq. (A1). The housing and storage portions vary by a different relationship, Eq. (A2) (Gyldenkærne et al., 2005):

$$E = T_e^{0.89} \times V^{0.26} \ , \tag{A2}$$

where ammonia emissions (E) incorporate effective temperature ($T_e$) and ventilation rate (V). Storage temperature ($T_e$) and

ventilation rate (V = w) are not species dependent, but housing $T_e$ and V do vary by species and their housing types. The relative weight of each manure emissions component (application, housing, and storage) is also species dependent (Velthof et al., 2012). Each month is scaled from the base year (2005) emissions in that month, and so the emissions changes depend on the meteorology of 2005. For example, the T in the Midwest in both summers 2005 and 2012 are similarly above the 10- and 35-year mean T from MERRA. This decreases the effect of volatilization on ammonia emissions in the Midwest in summer

2012 while using this method. These scale factors are then applied separately to the livestock and fertilizer portions of the anthropogenic ammonia inventory as appropriate. Resulting emission magnitude changes are shown in Fig. 11.

**Acknowledgments**

This work was supported by NOAA (NA12OAR4310064). We thank D. Ridley for assistance with surface site weighted sampling and gridding county-level data, F. Paulot for guidance with emissions scaling, S. McKeen for NEI-2005 emissions

sectors, A. van Donkelaar for NEI-2005 emissions gridding, J. Mao for implementation of FINN into GEOS-Chem, and the GEOS-Chem support staff and community for model documentation and issue resolution. IASI is a joint mission of





EUMETSAT and the Centre National d'Etudes Spatiales (CNES, France). C. Clerbaux is grateful to CNES for financial support. L. Clarisse is a research associate with the Belgian F.R.S.-FNRS.

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



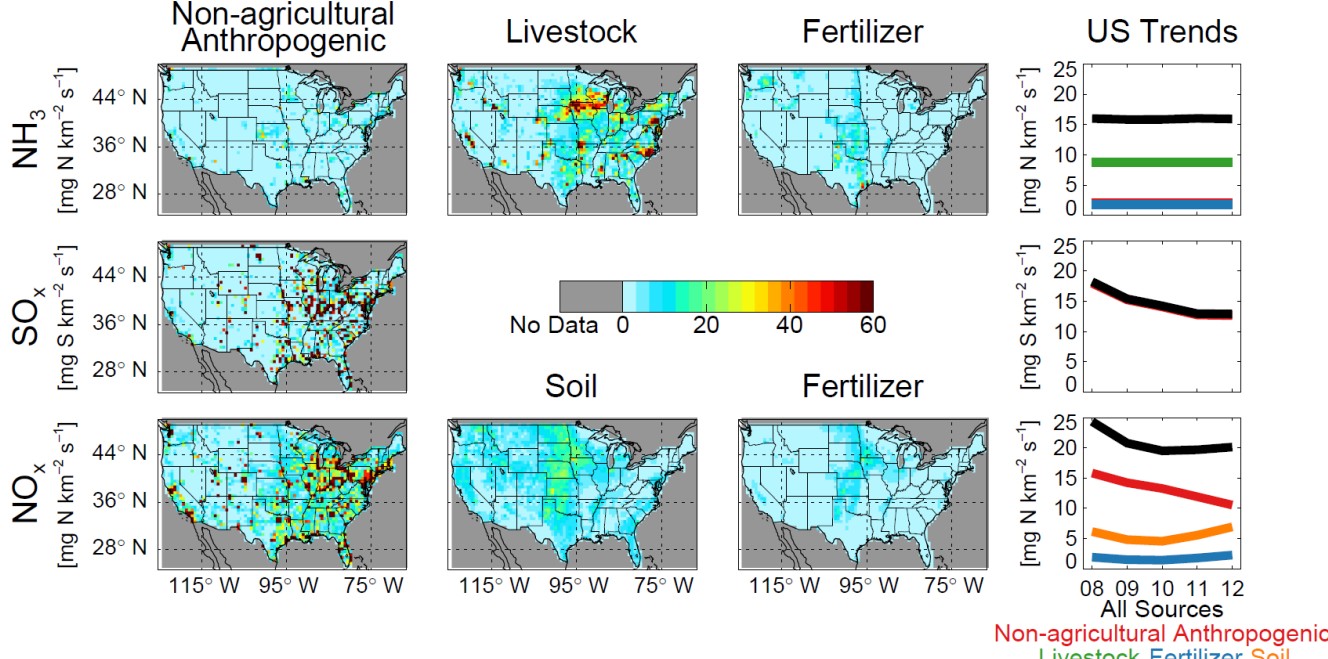

**Figure 1: Summer (JJA) ammonia (top row), SO$_x$ (middle row), and NO$_x$ (bottom row) emissions as implemented in the GEOS-Chem base scenario. Maps show values for 2008; US emission rate shown for 2008 through 2012 on the right. Color bar is saturated at 60; local values may exceed this emission rate. Data outside the continental US is not shown here nor in all subsequent figures.**



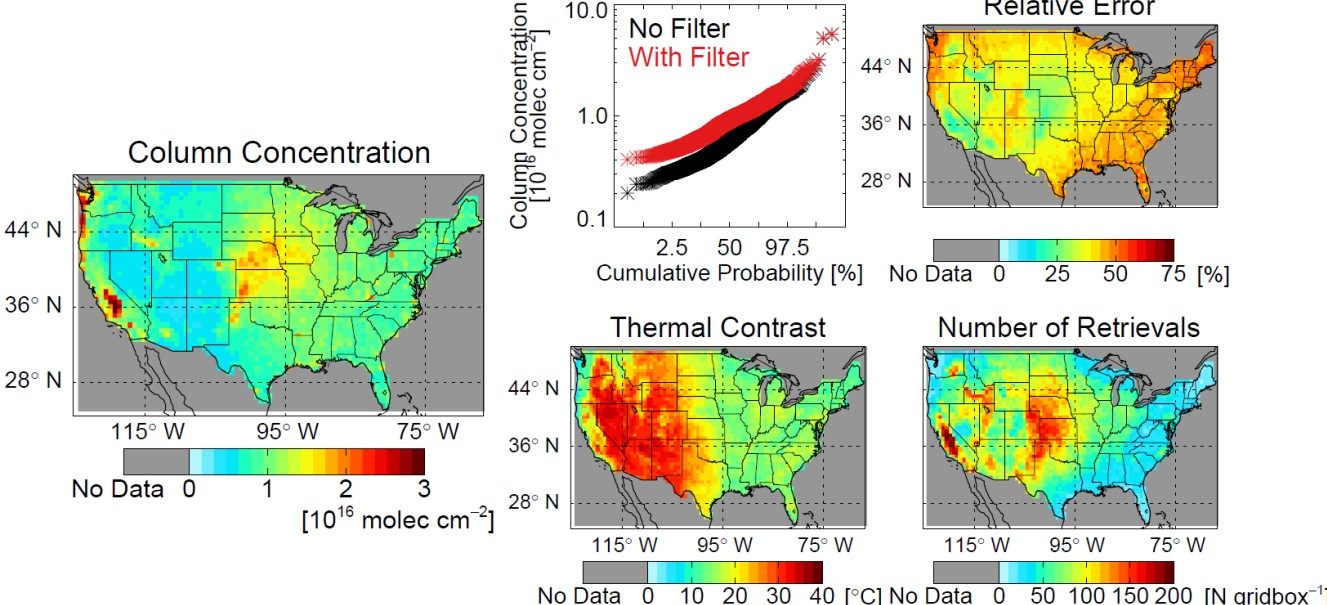

**Figure 2: Mean gridded daily summer (JJA) 2008–2012 IASI ammonia column concentrations (left), filtered for cloud cover (< 25 % cloud cover), skin temperature (> –10 °C), and relative error (≤ 75 %). Distribution of column concentrations with (red) and without (black) described filtering (top center). Accompanying retrieval parameters and properties: relative error (top right), thermal contrast (bottom center), and number of retrievals (bottom right).**



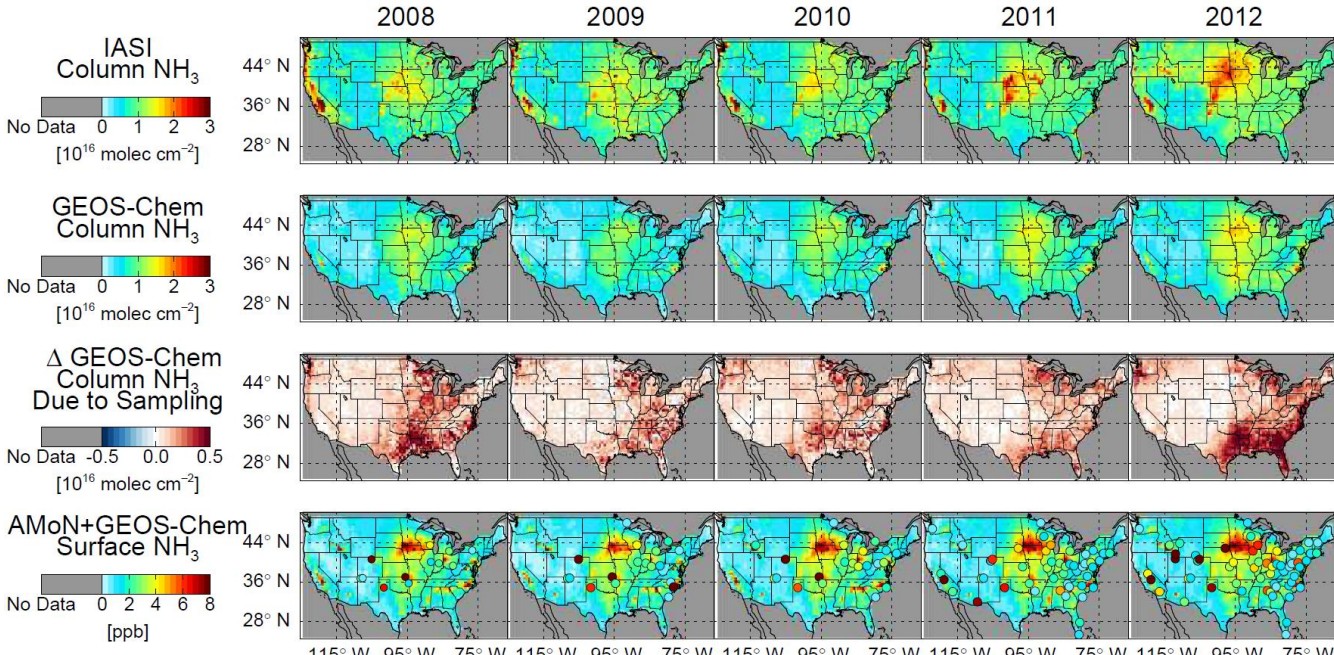

**Figure 3: Mean summer (JJA) ammonia concentrations for 2008 to 2012 (columns): gridded and filtered IASI observed column concentration, GEOS-Chem simulated column concentration sampled to valid IASI days, changes in GEOS-Chem simulated column concentration due to sampling to coincident IASI measurements, and AMoN observed surface concentration (circles) overlaid on GEOS-Chem surface concentration (rows, top to bottom).**

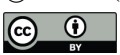

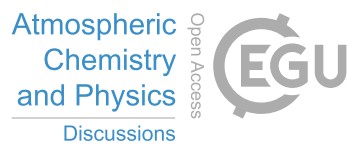
Atmospheric Chemistry and Physics Discussions — Open Access — EGU

**Table 1: Relevant details for the observed ammonia concentrations shown in Fig. 4, including the aircraft campaign, geographic location, dates, ammonia instrument, instrument sample rate, and typical instrument uncertainty range (calibration uncertainty + measurement imprecision), which show flight-to-flight variability detailed in the respective archived data files.**

| Campaign | Region (latitude; longitude range) | Dates | Instrument | Uncertainty |
|---|---|---|---|---|
| ICARTT (Nowak et al., 2007) | Northeastern US (38° N–47° N; 67° W–83° W) | Jul–Aug 2004 | NH$_3$ CIMS[a] | ± (30 % + 0.115 ppb) + 0.045 ppb |
| TexAQS (Nowak et al., 2010) | Eastern Texas (27° N–35° N; 90° W–100° W) | Sep–Oct 2006 | NH$_3$ CIMS | ± (25 % + 0.070 ppb) + 0.035 ppb |
| CalNex (Nowak et al., 2012) | Southern California (CA) (see Schiferl et al., 2014) | May–Jun 2010 | NH$_3$ CIMS | ± (30 % + 0.200 ppb) + 0.080 ppb |
| CalNex | Central Valley, California (see Schiferl et al., 2014) | May–Jun 2010 | NH$_3$ CIMS | ± (30 % + 0.200 ppb) + 0.080 ppb |
| DISCOVER-AQ | Central Valley, California (see Schiferl et al., 2014) | Jan–Feb 2013 | CRDS[b] (Picarro G2103) | ± (35 % + 1.7 ppb) + 0.2 ppb |
| SENEX | Southeastern US (31° N–43° N; 75° W–96° W) | Jun–Jul 2013 | NH$_3$ CIMS | ± (25 % + 0.070 ppb) + 0.020 ppb |
| FRAPPE | Colorado (38° N–42° N; 101° W–110° W) | Jul–Aug 2014 | Aerodyne Dual NH$_3$/HNO$_3$ QCL[c] | ± (22 % + 0.305 ppb) + 0.058 ppb |
| SONGNEX | Colorado (38° N–42° N; 101° W–110° W) | Mar–Apr 2015 | NH$_3$ CIMS | ± (35 % + 0.500 ppb) + 0.035 ppb |
| SONGNEX | Southern Plains (26° N–36° N; 93° W–105° W) | Mar–Apr 2015 | NH$_3$ CIMS | ± (35% + 0.500 ppb) + 0.035 ppb |

[a]CIMS: chemical ionization mass spectrometer, [b]CRDS: cavity ring down spectrometer, [c]QCL: quantum cascade laser



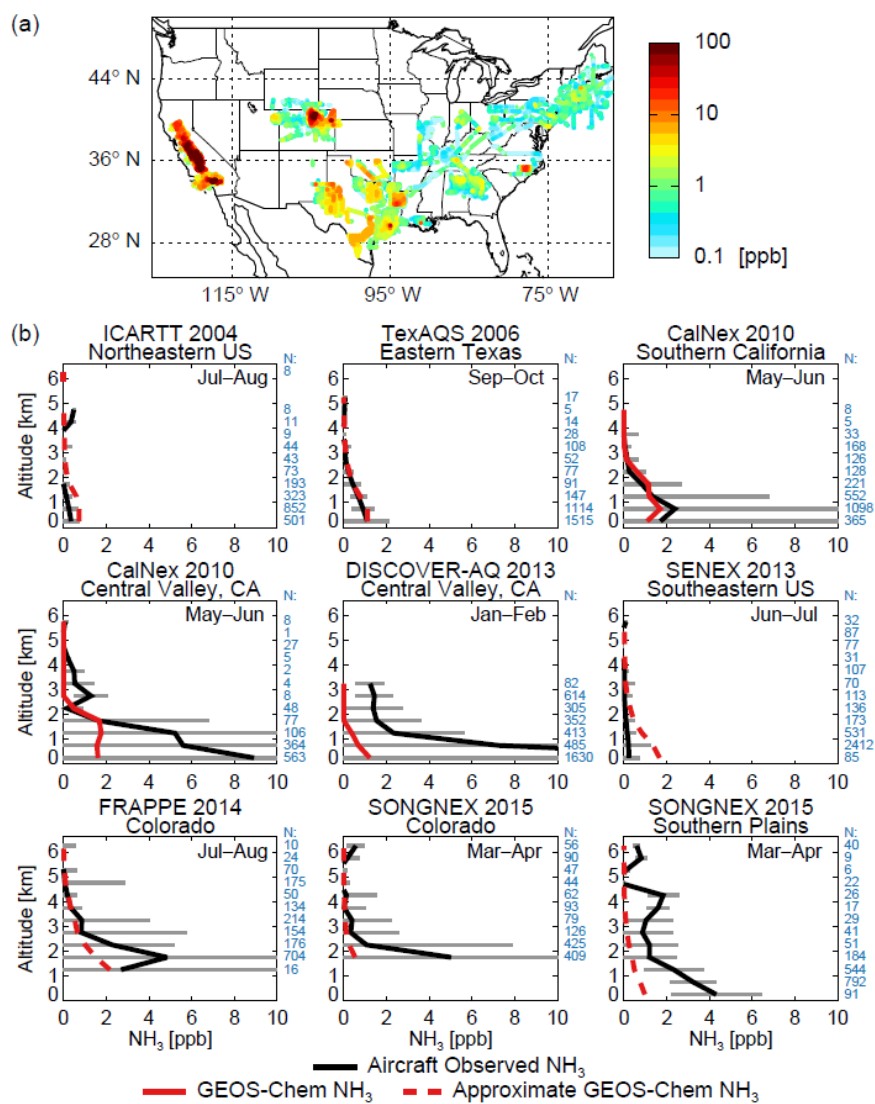

**Figure 4: (a) Spatial distribution of 1 min mean observed ammonia concentrations for several aircraft campaigns throughout the US listed in Table 1. (b) Vertical profiles of median observed ammonia concentration (black) and median GEOS-Chem simulated ammonia concentration (red) averaged in 500 m vertical bins from these campaigns. Simulated concentrations matched to the year and flight tracks of the campaign are shown in solid red, while approximately sampled concentrations (mean 2008–2012 simulated concentrations) are shown in dashed red. Gray bars show the standard deviation of observations in each bin. The number of observations in each bin are shown in blue. The two months during which the campaign took place is indicated in the top right of each profile.**





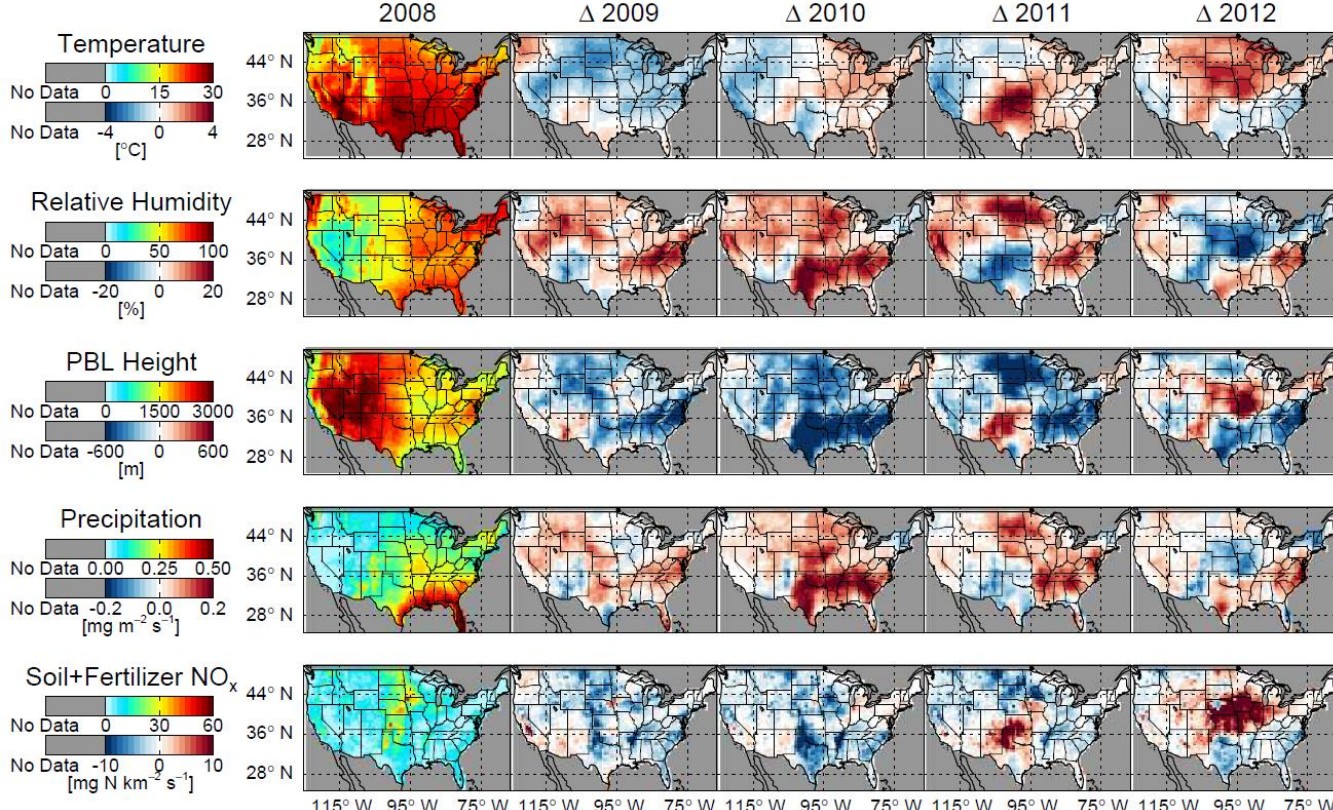

**Figure 5: Mean summer (JJA) assimilated GEOS-5 meteorology parameters and meteorologically driven NO$_x$ emissions used in GEOS-Chem simulation for 2008 to 2012 (columns): temperature, relative humidity, planetary boundary layer (PBL) height, precipitation, and soil + fertilizer NO$_x$ emissions (rows, top to bottom). Absolute values for 2008 shown along with changes from 2008 for 2009 to 2012.**





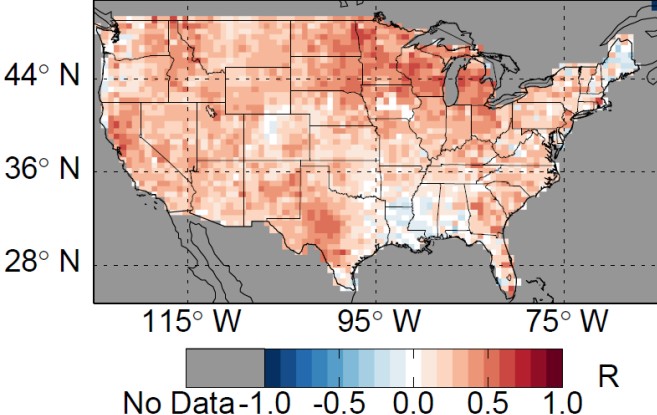

**Figure 6: Summer (JJA) correlation (R) for all years (2008–2012) between daily gridded and filtered IASI ammonia column concentration and daily GEOS-Chem base scenario ammonia column concentration.**





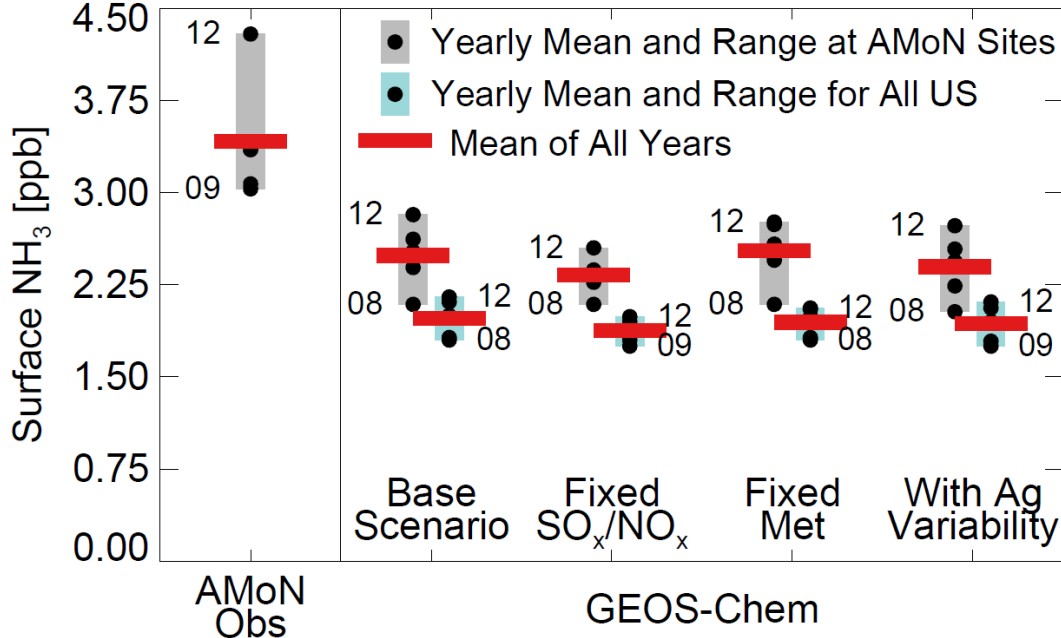

**Figure 7: Yearly summer (JJA) mean surface ammonia concentration (black circles) and mean of all years (red bar) for observed AMoN sites valid from 2008 to 2012 and four GEOS-Chem scenarios: base scenario, fixed anthropogenic SO$_x$ and NO$_x$ emissions, fixed meteorology, and including agriculture ammonia emission variability (left to right). Vertical bars indicate range of all years: simulation sampled to AMoN sites (gray) and simulation for entire US (blue).**





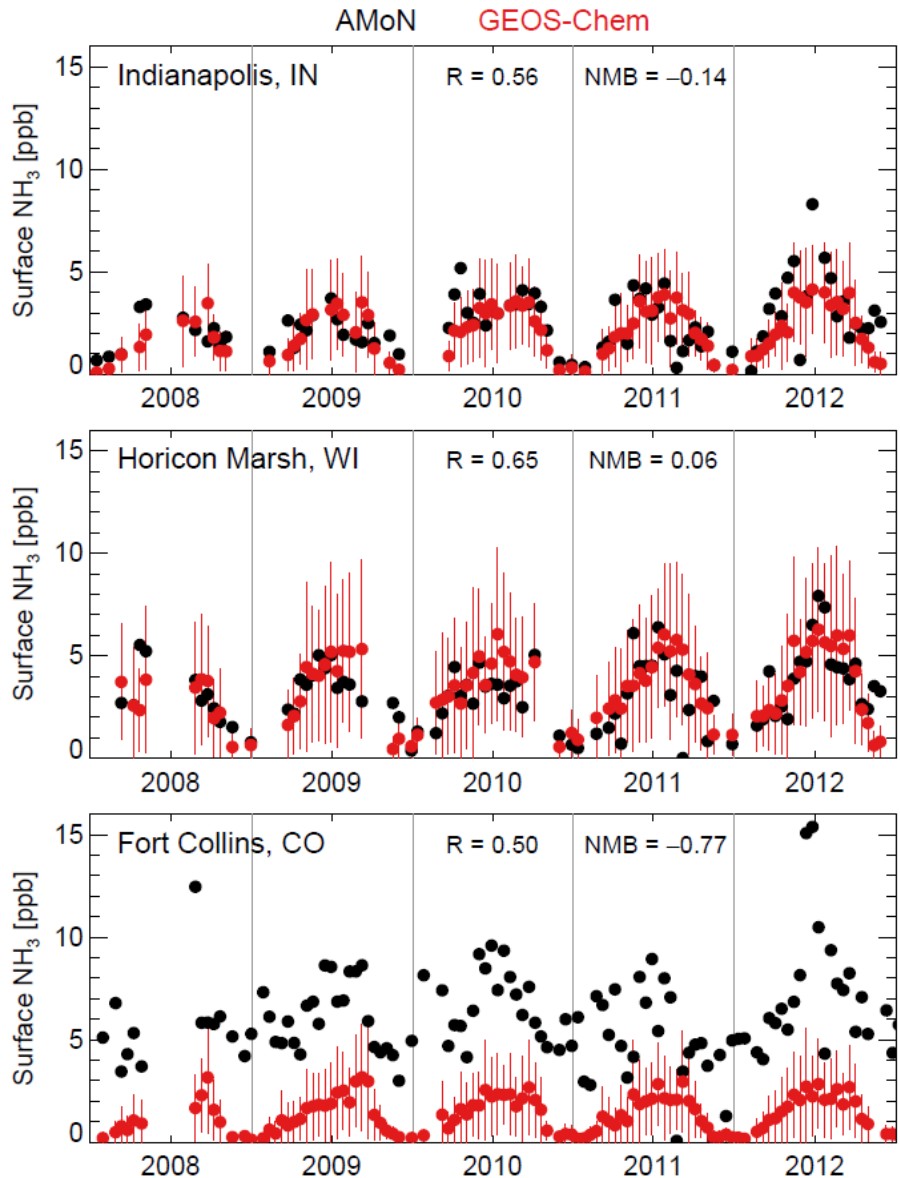

**Figure 8: Observed (black circles) and base scenario simulated (red circles) surface ammonia concentration time series at three AMoN sites from 2008 to 2012: Indianapolis, Indiana (IN) (top, urban), Horicon Marsh, Wisconsin (WI) (middle, agricultural), and Fort Collins, Colorado (CO) (bottom, varying topography / high horizontal gradient). Standard deviation of simulated hours shown as vertical red lines.**



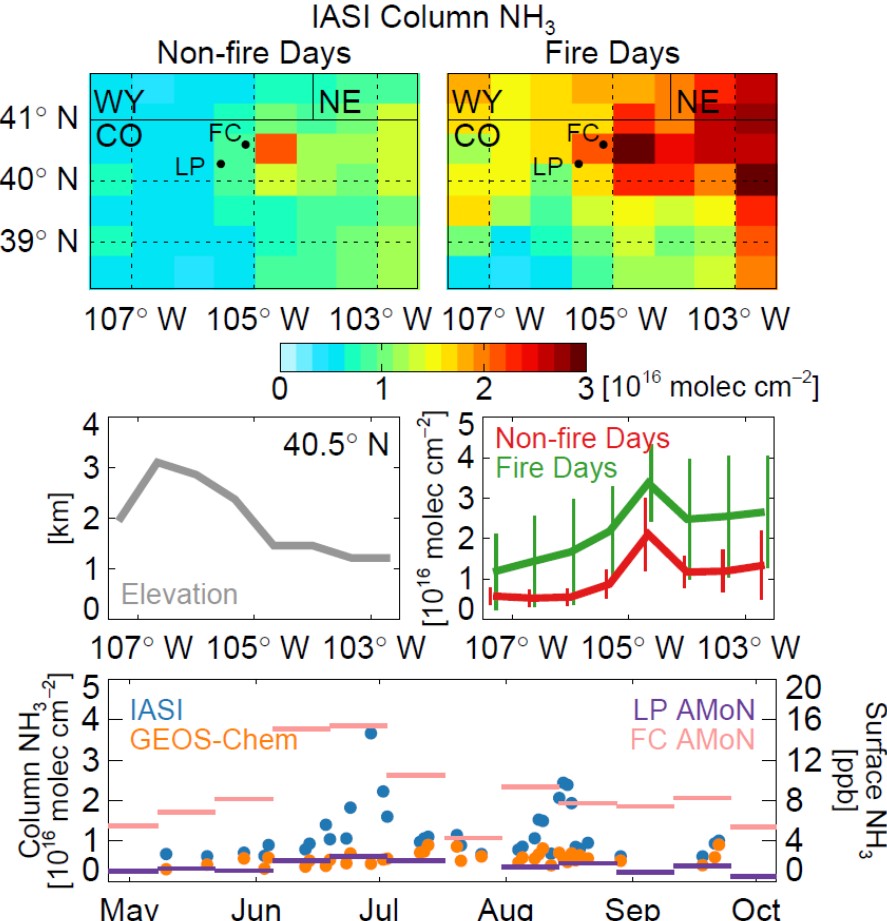

**Figure 9: Mean gridded IASI column ammonia concentration over the Colorado Front Range from May to September 2012 during non-fire days (top left) and fire days (top right). Elevation of each grid box center at 40.5° N over the longitude range above (middle left). Mean IASI column ammonia concentration at each grid box at 40.5° N over the longitude and time range above for non-fire days (red) and fire days (green) (middle right). Mean daily IASI (blue) and GEOS-Chem (orange) column ammonia concentrations over domain above and observed surface ammonia concentrations at Longs Peak (LP) (purple) and Fort Collins (FC) (pink) AMoN sites (bottom).**





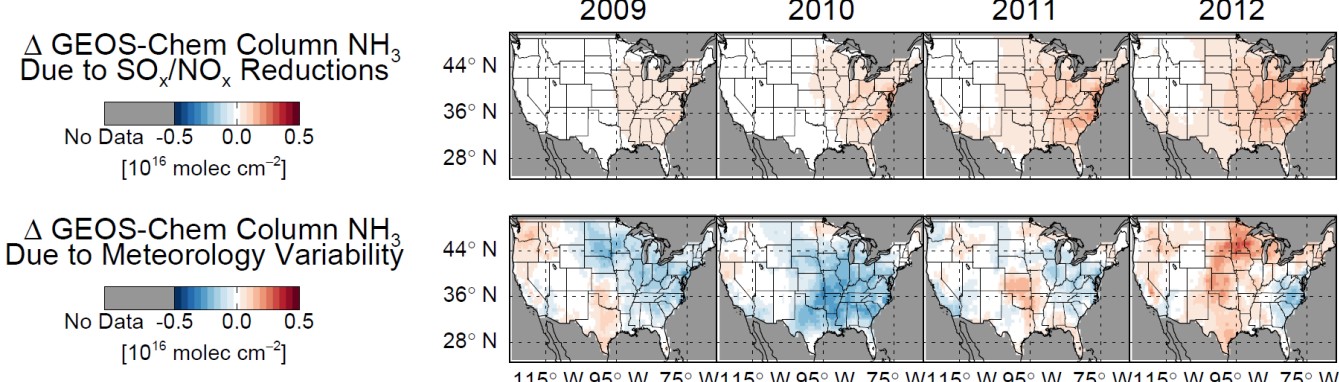

**Figure 10: Simulated mean summer (JJA) ammonia column concentrations changes for 2008 to 2012 (columns) caused by anthropogenic SO$_x$ and NO$_x$ emissions reductions and assimilated meteorology variability (rows, top to bottom). Compare to baseline ammonia column shown in Fig. 3.**





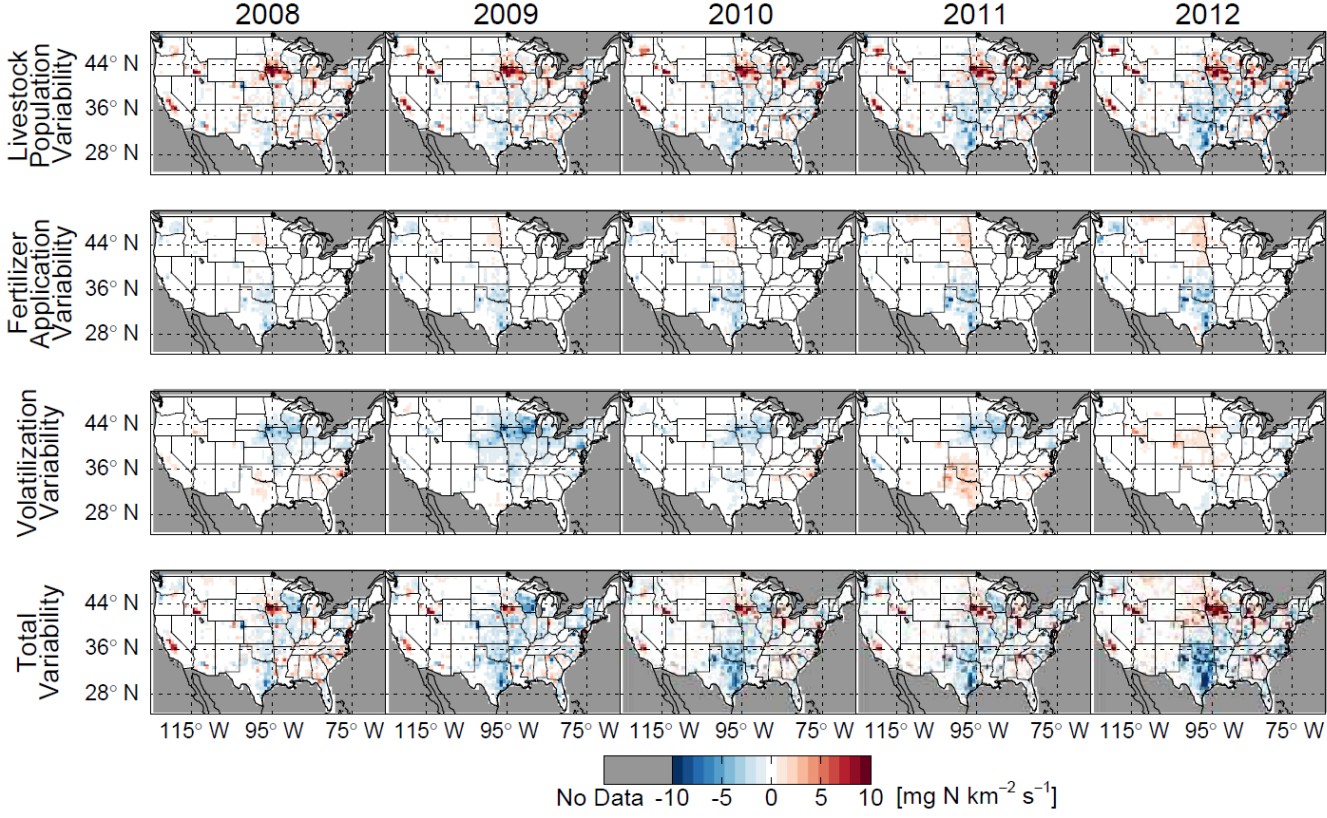

**Figure 11: Differences in summer (JJA) agricultural ammonia emissions compared to base scenario emissions for 2008 to 2012 (columns) by including various emissions variability scenarios: livestock population variability, fertilizer application variability, volatilization variability, and all three combined (rows, top to bottom).**





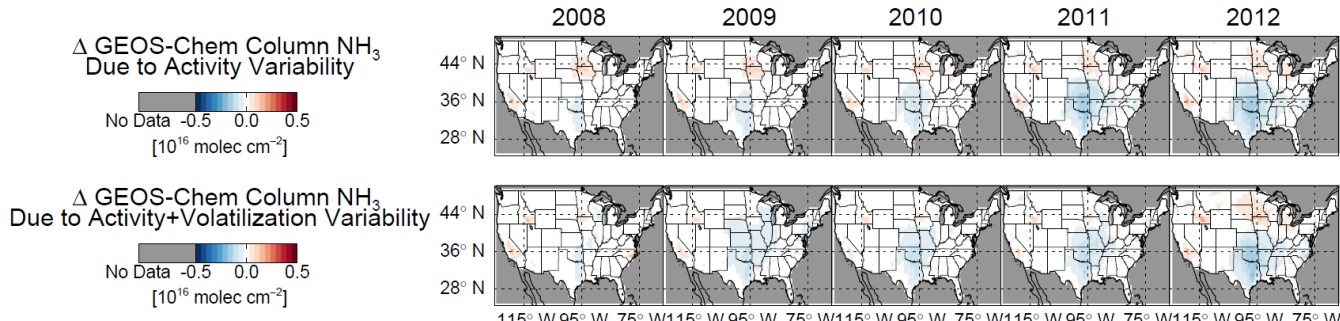

**Figure 12: Changes in simulated summer (JJA) surface ammonia concentration from base scenario caused by including variable ammonia emissions for 2008 to 2012 (columns): activity (livestock population and fertilizer application) variability and activity with volatilization variability (rows, top to bottom). Compare to baseline ammonia column shown in Fig. 3.**





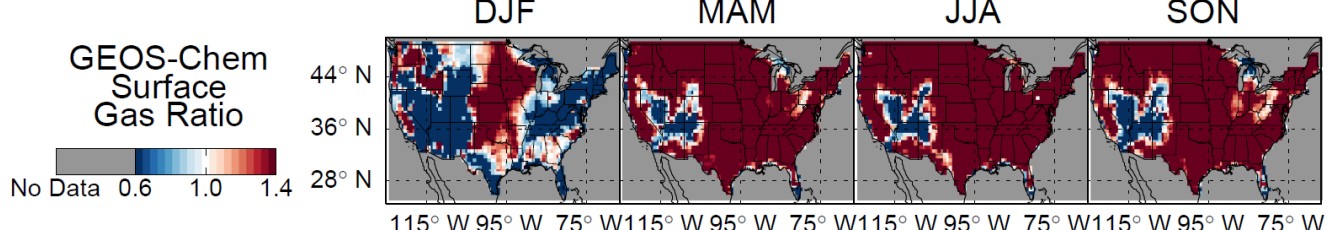

**Figure 13: Base scenario simulated mean seasonal gas ratio (GR) for all years (2008–2012). All values are greater than zero.**





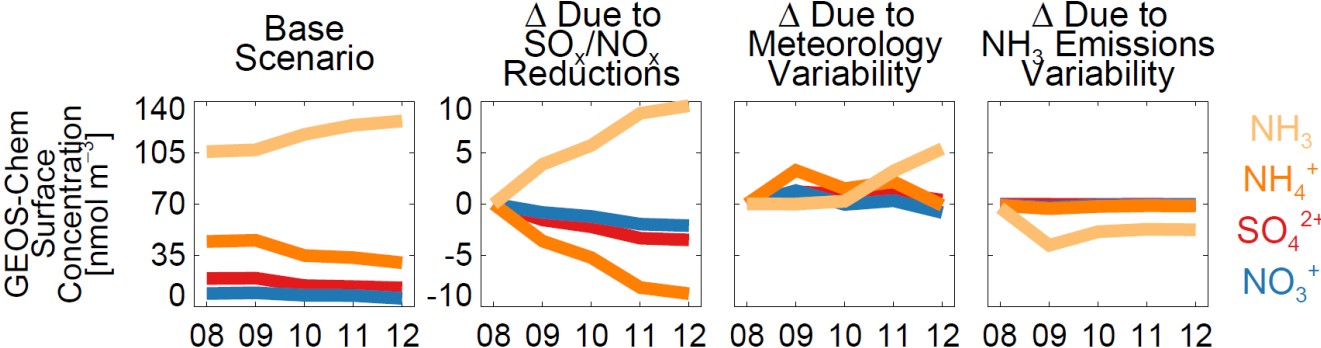

**Figure 14: Mean summer (JJA) surface concentrations over the US for relevant gas and particle species (ammonia gas (light orange), ammonium particle (dark orange), sulfate particle (red), and nitrate particle (blue)) for 2008 to 2012 for several scenarios: base scenario, changes from base scenario due to anthropogenic SO$_x$ and NO$_x$ emissions reductions, assimilated meteorology variability, and added agriculture ammonia emissions variability (left to right).**