# Peer review of "Interannual Variability of Ammonia Concentrations over the United States: Sources and Implications"

_Atmospheric Chemistry and Physics, 2016_

## Referee Comment (RC1) · Anonymous Referee #1 · 28 Jun 2016

Schiferl et al. use the GEOS-CHEM model to simulate ammonia concentrations across the U.S. during the period 2008-2012. They evaluate model performance by comparison with in situ (including AMoN) and IASI satellite measurements, use the model to examine factors driving interannual ammonia concentration variability, and infer from model simulations how emissions variability affects PM2.5 formation and reactive nitrogen deposition. While not entirely novel, the manuscript builds upon earlier model-observation intercomparisons and provides a more comprehensive look especially at factors influencing year-to-year variability in ammonia concentrations. The article is generally well written and clearly describes approach, hypotheses, and findings. There are several items that should be addressed to improve the manuscript:

1. In the abstract (line 20, p. 1), the authors should be more quantitative in summarizing the model bias in simulating observed ammonia concentrations.

2. p. 2, line 13: the relevant parameter here is sulfate, not SOx. An environment can have lots of SO2 without affecting ammonium nitrate formation, for example if there is insufficient oxidant to promote reaction of SO2 to sulfate. It is only when that SO2 is oxidized to sulfate that there is an effect.

3. The authors could make a stronger case for the importance of their work by increasing emphasis in the introduction on the increasingly important contributions ammonia/ammonium are making to reactive N deposition. While U.S. NOx reductions are reducing oxidized nitrogen deposition, U.S. ammonium deposition has been increasing. The source of this increase is not well understood. The work presented here could help lay a foundation for better future understanding that change.

4. The paper's focus primarily on summer ammonia is somewhat disappointing. While usable satellite obs are more limited in other seasons, some of the most interesting effects for PM are, as the authors point out, at cooler times of year. The focus on August emissions in Fig. 1 is also a little disconcerting. Major fertilizer emissions from the U.S. heartland (Iowa, Illinois, Indiana, etc...) are missing from this figure because they occur in spring, for example, while this figure just shows August. At a minimum, some additional discussion about how emissions change regionally throughout the year would be helpful to the reader.

5. Considerable attention has been focused in the past couple years on improved treatments of ammonia deposition in models, especially through the incorporation of bidirectional flux schemes. It is surprising that this issue is not mentioned at all in this manuscript, particularly since such "bidi" treatments tend to reduce ammonia deposition, especially near sources, helping models better match observations. At a minimum the authors should discuss the deposition scheme used in their simulations and outline shortcomings of the treatment for ammonia dry deposition.

6. The column concentrations in Fig. 2 show a surprising depiction of very high values along the coastal region of the Pacific Northwest. What do the authors think of this? This is not a region typically identified with high ammonia concentrations and I note that the number of retrievals here is fairly small. Do the authors think this depiction is realistic?

7. p. 11, line 14: Should the 2.8 ppb mentioned here be the 2.5 ppb mean simulated at AMoN sites?

8. p. 11, line 18: I would be reluctant to directly state that the "AMoN network as a whole has a sampling bias." The goal of the AMoN network is not a statistical sampling to obtain the average U.S. concentration. Please qualify your statement so that someone does not quote you out of context.

9. The 3-site comparison with the model nicely illustrates some of the characteristics of regions where the model predicts ammonia concentrations better or worse. Very nice. I suggest adding plots with similar comparisons for your other sites to supplemental information as many readers might find this useful.

10. p. 13, lines 18-19. It is unclear how you know the LP AMoN site is not showing fire influence. Certainly this site is influenced by upslope transport from the Front Range region for some hours almost every day during the summer.

11. I found myself wondering while reading why you used NEI 2005 rather than NEI 2011 ammonia emissions as your base case. I wondered this even more after seeing in section 4.4 where you compared the two and found NEI 2011 offered some improvements. Please justify.

12. Section 5.1 on effects of SO2 and NOx emissions reductions is very interesting, but you need to discuss whether the model performance for sulfate and nitrate/nitric acid is good enough to reliably interpret the findings to such precision (e.g., change of 32%/0.17 ppb).

13. The discussion of meteorological variability effects in section 5.2 is also interesting, but could be strengthened by examining the model phase partitioning to support your arguments about the effect of that phenomenon on ammonia concentrations.

14. p. 17, lines 25-27. The sentence beginning "Nearly all changes to the ammonia concentrations..." is confusing since it might make the reader think the volatilization scaling effect is affecting phase partitioning in the atmosphere rather than changes in ammonia volatilization from sources.

15. Please specify that the concentrations in equation 1 are in molar units for clarity.

16. The discussion of the Gas Ratio (GR) on p. 19 needs to be revised. First, the statement that GR>1 implies little potential for further ammonium nitrate formation is an oversimplification. Yes, this is true for GR »1, but the dropoff in effect is more gradual than implied since the equilibrium concentration of ammonium nitrate is proportional to the product of the ammonia and nitric acid concentrations. This product can still grow as GR increases above 1 but will start to taper off for much larger values. By choosing GR =1 as a de facto cutoff for regions/times where ammonium nitrate formation may respond to ammonia concentrations you are oversimplifying what is really a more gradual change. Given how important this issue is to considering future policy re: ammonia emissions, I think you should be more careful in how you describe this effect.

17. While ammonium nitrate formation is limited by warm summertime temperatures, it would be worth mentioning that it might be important overnight as T drops and RH rises and that other ammonium salts (e.g., ammonium oxalate) might be important during this more photochemically active time of year.

18. The authors do a nice job summarizing the needs for better observational constraints on ammonia concentrations in the future. I would add that such constraints should also feature (1) higher time resolution measurements and (2) measurements of both gas and particle phase ammonia/ammonium to (1) provide a better basis for comparison with model simulations with reduced variability in meteorological conditions

and source impact and (2) to better constrain the total NHx budget.

19. The x-axis timelines in Figure 8 should be better labeled/identified.

---

## Referee Comment (RC2) · Anonymous Referee #2 · 18 Jul 2016

The article title, "Interannual Variability of Ammonia Concentrations over the United States: Sources and Implications", by Schiferl et al., is well written and timely given the limited knowledge of ammonia variability. Provided below are some review comments.

General Comments:

1) The paper talks about changes in the transfer of ammonia from the surface to the atmosphere due to temperature and windspeed (volatilization scaling), but does not put it in the context of bi-direction exchange and gas-aerosol phase transitions. The deposition and re-emission processes in the bidirectional exchange extends the spatial range of influence of the NH3 emissions, and hence the NH3 lifetime (e.g. Zhu et al., 2015). NH3 also contributes to the formation of atmospheric aerosols that can

reside and be transported in the atmosphere for several days to a week releasing NH3 back into the atmosphere modifying the variability of ammonia concentrations. It would be good if the authors could provide insights on the impacts of the variability due to bi-directional flux. Zhu L., D.K. Henze, J.O. Bash, G.-R. Jeong, K.E. Cady-Pereira, M.W. Shephard, M. Luo, F. Paulot, and S. Capps, Global Evaluation of Ammonia Bi-Directional Exchange and Livestock Diurnal Variation Schemes, Atmos. Chem. Phys, 15, 12823-12843, doi:10.5194/acp-15-12823-2015, 2015.

2) The paper recognizes the limitation of the satellite observations due to lack of vertical information. It would be good to note that this is not general to satellite observations, but the particular IASI ammonia retrieval algorithm used in the study. For example, the new CrIS NH3 optimal estimation retrievals (Shephard et al., 2015) will be able to provide this type of information (e.g. averaging kernels and covariance matrics) allowing for more quantitative comparisons against the model simulations.

3) It is still not totally clear how the impact of the spatial sampling between the model and the observations impact the measurement variability. For a study over just North America, why was a global GEOS-Chem model used instead of a more regional model (i.e. CMAQ) to investigate the ammonia variability? A regional model would at least have a spatial sampling that is more representative for comparisons with the observations.

Minor Comments:

1) Page 5, line 27. Also should add in AIRS and CrIS.

2) Page5, lines 29-30. "...calculated from a wider spectral range than previous ammonia products,...". It is not clear if the point is to just state this fact, or imply that this is better. Using a wider spectral coverage does not necessarily produce a better retrieved product. For example, a robust spectral window selection approach can be based on the maximum information content by taking into consideration errors (e.g. interfering species, spectroscopic errors, measurement errors, etc.) (e.g. Echle et al., (2000) and

Worden et al. (2004)).

Echle, G., T. von Clarmann, A. Dudhia, J. M. Flaud, B. Funke, N. Glatthor, B. Kerridge, M. Lopez-Puertas, F. J. Martin-Torres, and G. P. Stiller (2000), Optimized spectral microwindows for data analysis of the Michelson Interferometer for Passive Atmospheric Sounding on the environmental satellite, Appl. Opt., 39(30), 5531–5540.

Worden J, S. Sund, M.W Shephard, S.A Clough, H. Worden, K Bowman, A Goldman. Predicted errors of tropospheric emission spectrometer nadir retrievals from spectral window selection. J Geophys Res. 2004;109:doi:10.1029/2004JD004522.

3) Page 6, line 1: Please state what forward radiative transfer model was used.

4) Page 6, line 5: Are these uncertainties relative, or absolute, or both?

5) Page 6, line 14: remove "present"

6) Page 6, line 16: Maybe also add to the line ending in "...distributed measurements" the additional "and the differences in measured quantities.", which leads nicely into the next sentence.

7) Page 9, lines 25-29: should mention in addition to vertical sensitivity, the last of the actual information content limits the comparison.

8) Page 10, lines 10-15. Could the lack of variability also be due to the fact that satellite total column values are being used, rather than information from only the parts of the profile where the satellite is sensitive (e.g. often limited information right at the surface).

---

## Author Comment (AC1) · 12 Sep 2016

Response to Referee #1

Note: page and line references mentioned in author changes refer to positions within the revised manuscript below.

1. In the abstract (line 20, p. 1), the authors should be more quantitative in summarizing the model bias in simulating observed ammonia concentrations.

**To address this we have added "(by 26 % at surface sites)" in page 1, line 21.**

2. p. 2, line 13: the relevant parameter here is sulfate, not SOx. An environment can have lots of SO2 without affecting ammonium nitrate formation, for example if there is insufficient oxidant to promote reaction of SO2 to sulfate. It is only when that SO2 is oxidized to sulfate that there is an effect.

**We agree and have clarified this in the text.**

**Changed "SOx" to "sulfate" and modified sulfate definition in page 2, line 13; Defined SO2 in page 2, line 15; Defined SOx in page 2, line 22.**

3. The authors could make a stronger case for the importance of their work by increasing emphasis in the introduction on the increasingly important contributions ammonia/ammonium are making to reactive N deposition. While U.S. NOx reductions are reducing oxidized nitrogen deposition, U.S. ammonium deposition has been increasing. The source of this increase is not well understood. The work presented here could help lay a foundation for better future understanding that change.

**Thank you to the referee for this suggestion. We have added a sentence on this to the Introduction.**

**Added "…the proportion of reactive nitrogen deposition is shifting from oxidized to reduced forms (Pinder et al., 2011; Lloret and Valiela, 2016), and thus…" starting in page 3, line 3.**

4. The paper's focus primarily on summer ammonia is somewhat disappointing. While usable satellite obs are more limited in other seasons, some of the most interesting effects for PM are, as the authors point out, at cooler times of year. The focus on August emissions in Fig. 1 is also a little disconcerting. Major fertilizer emissions from the U.S. heartland (Iowa, Illinois, Indiana, etc...) are missing from this figure because they occur in spring, for example, while this figure just shows August. At a minimum, some additional discussion about how emissions change regionally throughout the year would be helpful to the reader.

**We agree with the referee that some of the interesting effects for PM are at cooler times of years, and while we discuss those briefly, we focus on summer as this is the season with strong satellite constraints. We hope that future studies can explore the spring/fall time period in greater detail with new datasets.**

**To clarify: we focus on August emissions as this is the base provided by NEI-2005; all other monthly values are scaled to August. The failure to capture fertilizer emissions in spring is therefore an inherent weakness of the ammonia emissions provided by NEI-2005. This is mentioned in Sects. 2.2 and A1. We have added additional clarification in the main text.**

**Added to page 4, line 21: "This proportion is unrealistically constant throughout the year as the scaling above does not, for example, account for springtime crop fertilization."**

5. Considerable attention has been focused in the past couple years on improved treatments of ammonia deposition in models, especially through the incorporation of bidirectional flux schemes. It is surprising that this issue is not mentioned at all in this manuscript, particularly since such "bidi" treatments tend to reduce ammonia deposition, especially near sources, helping models better match observations. At a minimum the authors should discuss the deposition scheme used in their simulations and outline shortcomings of the treatment for ammonia dry deposition.

**We thank the referee for pointing out this issue. We have added discussion of the current Wesley deposition scheme in our model and acknowledge the potential weaknesses in this unidirectional scheme and share advancements made recently in simulating bi-directional flux. However, we note that implementation of "bidi" into GEOS-Chem by Zhu et al. (2015) does not show uniform improvement compared to limited observations. Such schemes are challenging to implement and evaluate in large-scale models such as ours due to fine-scale variability in local conditions which may affect the calculated compensation point.**

**We have added discussion beginning on page 5, line 9: "We have not included any scheme which accounts for the bidirectional flux (deposition and re-emission) of ammonia in our base scenario. Rather, ammonia is permanently removed via wet scavenging in convective and stratiform precipitation (Mari et al., 2000; Amos et al., 2012) and via surface resistance-driven dry deposition (Wesley, 1989). Ongoing research suggests that a unidirectional dry deposition scheme may be inappropriate with regards to ammonia (Massad et al., 2010; Zhang et al., 2010). Under a bidirectional scheme, ammonia can be either taken-up or re-emitted by a plant based on the comparison of the ambient ammonia concentration with a varying compensation point (an ambient concentration greater than the compensation point leads to deposition). Re-emitted ammonia has the potential to affect ecosystems farther downwind. Failing to account for this re-emission may locally cause an overestimation in dry deposition resulting in low ammonia concentrations. Zhu et al. (2015) incorporate the bidirectional flux scheme of Pleim et al. (2013) into GEOS-Chem, which increases the July ammonia emissions and concentration in the US. This slightly reduces the July model bias compared to measurements at Ammonia Monitoring Network (AMoN)**

**sites. However, the bidirectional scheme causes a decrease in ammonia emissions and concentration in April and October, which worsens the comparison with observations and does not account for missing primary emissions. Such bidirectional flux schemes, developed largely to simulate field conditions, require higher resolution observations for evaluation at finer scales than those offered by current observations and global models."**

6. The column concentrations in Fig. 2 show a surprising depiction of very high values along the coastal region of the Pacific Northwest. What do the authors think of this? This is not a region typically identified with high ammonia concentrations and I note that the number of retrievals here is fairly small. Do the authors think this depiction is realistic?

**We thank the referee for pointing this out. The high values along the Pacific Northwest coast are due to retrievals of ammonia over the ocean being incorporated into our land-based analysis. These ocean values tend to be much higher than those nearby over land. Since our ocean grid box mask is applied after gridding the retrievals to match the model resolution, some grid boxes such as those along the Pacific Northwest coast may contain up to 50% ocean. We have clarified this in the text.**

**Added to page 7, line 13: "We also isolate the continental US by removing grid boxes over Canada, Mexico and the ocean, but due to their size, some grid boxes along the border may exhibit outside influence (such as ocean retrievals along the Pacific Northwest coast)."**

7. p. 11, line 14: Should the 2.8 ppb mentioned here be the 2.5 ppb mean simulated at AMoN sites?

**Corrected. In page 12, line 6: change "2.8 ppb" to "2.5 ppb".**

8. p. 11, line 18: I would be reluctant to directly state that the "AMoN network as a whole has a sampling bias." The goal of the AMoN network is not a statistical sampling to obtain the average U.S. concentration. Please qualify your statement so that someone does not quote you out of context.

**We have qualified this statement as suggested by the referee.**

**Changed the sentence beginning in page 12, line 10 to: "This suggests that the AMoN network does not adequately represent the range of ammonia concentrations across the US; as many AMoN sites are located near high ammonia source regions, there is a sampling bias for this network."**

9. The 3-site comparison with the model nicely illustrates some of the characteristics of regions where the model predicts ammonia concentrations better or worse. Very nice. I suggest adding

plots with similar comparisons for your other sites to supplemental information as many readers might find this useful.

**We have added comparisons for the eight remaining sites to supplemental information.**

**Added Figs. S1-S3 to supplemental information. Added to page 13, line 17: "Similar comparisons for the eight remaining sites with records during this time period are shown in Figs. S1-S3."**

10. p. 13, lines 18-19. It is unclear how you know the LP AMoN site is not showing fire influence. Certainly this site is influenced by upslope transport from the Front Range region for some hours almost every day during the summer.

**We do not exclude the possibility of upslope transport, but the flat concentrations do not support any large fire enhancement. We have modified the text to highlight this point.**

**In page 14, line 14, change "…show no evidence of fire influence as the site is isolated…" to "…show no evidence of an enhancement due to fire, likely because the site is isolated…"**

11. I found myself wondering while reading why you used NEI 2005 rather than NEI 2011 ammonia emissions as your base case. I wondered this even more after seeing in section 4.4 where you compared the two and found NEI 2011 offered some improvements. Please justify.

**This research was started prior to the availability of NEI 2011. We included Sec. 4.4 because we anticipated the referee's question and expect that many readers might raise this issue. However, as we show, NEI 2011 does not offer an overall improvement (spring fertilizer maximum too high, for example) in the model simulation, and therefore we did not see a strong motivation for repeating our entire analysis with this inventory.**

**We clarify that the comparison may "worsen" in page 14, line 31.**

12. Section 5.1 on effects of SO2 and NOx emissions reductions is very interesting, but you need to discuss whether the model performance for sulfate and nitrate/nitric acid is good enough to reliably interpret the findings to such precision (e.g., change of 32%/0.17 ppb).

**The referee raises a good point and we acknowledge the importance of addressing the model performance in simulating sulfate and nitrate. As such, we have undertaken a comparison of the base scenario with IMPROVE network observations and added discussion on this to the text. The main relevant criteria is the model simulation of the trend. As the model reproduces the trend well (less so for nitrate) over areas with high ammonia concentrations, we justify using such precision to describe our findings.**

**Added starting in page 15, line 17: "This analysis relies on an accurate simulation of the trends in sulfate and nitrate in areas of significant ammonia concentration. Briefly, we evaluate our base scenario against observations from all available sites (148) in the Interagency Monitoring of Protected Visual Environment (IMPROVE) network (vista.cira.colostate.edu/improve) over our study period. Comparison of the trend in summer mean indicates that GEOS-Chem reproduces well the decreasing trend in sulfate over the eastern US and the Pacific Coast (not shown). In the Intermountain West, which generally lacks high ammonia concentrations, the simulation predicts a decreasing trend in sulfate, while the observations show an increase. The model generally reproduces the trend in nitrate, although the decline in nitrate in the Eastern US is somewhat stronger than observed. This indicates a possible over-sensitivity to changing $NO_x$ emissions in the model."**

13. The discussion of meteorological variability effects in section 5.2 is also interesting, but could be strengthened by examining the model phase partitioning to support your arguments about the effect of that phenomenon on ammonia concentrations.

**We thank the referee for this suggestion and perform an additional sensitivity simulation isolating the effects of partitioning. Results and discussion have been added to the text and figures to show the effected of partitioning variability.**

**We added a third row to Fig. 10 which shows the effects of partitioning variability and modified the caption as needed. Added to page 17, line 1: "A third sensitivity simulation isolates the effects of two-way partitioning of ammonia on the simulated ammonia concentration. This partitioning is driven by the ambient temperature and relative humidity as inputs into ISOROPPIA II. In this simulation, we hold these inputs constant at year 2008 conditions for all years of our simulation (2008–2012). Higher temperature and lower relative humidity generally favors partitioning into the gas phase and an increase in ammonia concentration. The results of this simulation, shown in Figure 10, indicate that the effects of partitioning are less spatially and temporally variable than those of all meteorology discussed above. The variability due to partitioning can make up a significant portion of the change due to all meteorology, such as in the warm summer of 2012 when partitioning accounts for 73 % of the net change due to all meteorology. This is also true to a smaller degree during the cool summer of 2009 (13%). In relatively wet summers, such as 2010 and 2011, enhanced partitioning acts to offset the losses due to all meteorology (likely caused by increased wet deposition) by 10 % and 73 % respectively. Overall, partitioning accounts for 23 % ($0.06 \times 10^{16}$ molec cm$^{-2}$) of the range in the summer base scenario column concentrations, which is 33 % of the range due to all meteorology. Thus, the phase partitioning due to meteorology plays a significant, but not always dominant, role in controlling the variability of ammonia."**

14. p. 17, lines 25-27. The sentence beginning "Nearly all changes to the ammonia concentrations..." is confusing since it might make the reader think the volatilization scaling

effect is affecting phase partitioning in the atmosphere rather than changes in ammonia volatilization from sources.

**We agree that this is confusing and have edited for clarity.**

**In page 19, line 5: Change "Nearly all changes to the ammonia concentrations follow directly from changes in the ammonia emissions since summer meteorology generally favors the gas phase of the ammonium nitrate equilibrium." to "Since summer meteorology generally favors the gas phase of the ammonium nitrate equilibrium, most ammonia resides in the gas phase, and nearly all changes to the ammonia concentrations in our scenarios correspond directly with changes in the ammonia emissions."**

15. Please specify that the concentrations in equation 1 are in molar units for clarity.

**Added in page 20, line 12: "The concentration in Eq. (1) are in molar units."**

16. The discussion of the Gas Ratio (GR) on p. 19 needs to be revised. First, the statement that GR>1 implies little potential for further ammonium nitrate formation is an oversimplification. Yes, this is true for GR »1, but the dropoff in effect is more gradual than implied since the equilibrium concentration of ammonium nitrate is proportional to the product of the ammonia and nitric acid concentrations. This product can still grow as GR increases above 1 but will start to taper off for much larger values. By choosing GR =1 as a de facto cutoff for regions/times where ammonium nitrate formation may respond to ammonia concentrations you are oversimplifying what is really a more gradual change. Given how important this issue is to considering future policy re: ammonia emissions, I think you should be more careful in how you describe this effect.

**We agree that this transition is gradual and have clarified our discussion.**

**In page 20, line 13: change "GR > 1" to "GR >> 1", in line 14: add "generally", and in line 16 add "We recognize that the transition around GR = 1 occurs gradually as ammonia increases, but note that a large portion of the US exhibits a GR well above or below 1 in all seasons."**

17. While ammonium nitrate formation is limited by warm summertime temperatures, it would be worth mentioning that it might be important overnight as T drops and RH rises and that other ammonium salts (e.g., ammonium oxalate) might be important during this more photochemically active time of year.

**We thank the referee for the suggestions and have added this to the text.**

**Added in page 20, line 25: "Although not evaluated here, summertime PM2.5 may be affected during overnight periods when temperature decreases and relative humidity**

**increases and via formation of minor salts such as ammonium oxalate, which are more likely to form during periods of high photochemistry."**

18. The authors do a nice job summarizing the needs for better observational constraints on ammonia concentrations in the future. I would add that such constraints should also feature (1) higher time resolution measurements and (2) measurements of both gas and particle phase ammonia/ammonium to (1) provide a better basis for comparison with model simulations with reduced variability in meteorological conditions and source impact and (2) to better constrain the total NHx budget.

**These are good suggestions; we have added them to the conclusions.**

**We added in page 22, line 28: "Future surface monitoring sites should be distributed across source and background regions, make higher temporal resolution measurements, and measure both gas and particle phase NHx. This will reduce the variability due to meteorology and source condition, shown in our study to be large, and better constrain the entire NHx budget."**

19. The x-axis timelines in Figure 8 should be better labeled/identified.

**In Fig. 8, we added "Time" label to the x-axis, increased size of fonts relative to plot, removed tick marks on x-axis, and added to caption "
[revised manuscript text omitted]

**Supplemental Figures**

[Figure]

Figure S1: Same as Fig. 8, but for: Detroit, Michigan (MI) (top, urban), Athens, Ohio (OH) (middle, mixed forest / agricultural), and Ithaca, New York (NY) (bottom, mixed forest / agricultural).

[Figure]

**Figure S2: Same as Fig. 8, but for: Canonceta, Texas (TX) (top, agricultural), Stilwell, Oklahoma (OK) (middle, agricultural), and Bondville, Illinois (IL) (bottom, agricultural).**

[Figure]

**Figure S3: Same as Fig. 8, but for: Farmington, New Mexico (NM) (top, varying topography / high horizontal gradient) and Navajo Lake, NM (bottom, varying topography / high horizontal gradient).**

---

## Author Comment (AC2) · 12 Sep 2016

**Response to Referee #2**

Note: page and line references mentioned in author changes refer to positions within the revised manuscript below.

**General Comments:**

1) The paper talks about changes in the transfer of ammonia from the surface to the atmosphere due to temperature and windspeed (volatilization scaling), but does not put it in the context of bidirection exchange and gas-aerosol phase transitions. The deposition and re-emission processes in the bidirectional exchange extends the spatial range of influence of the NH3 emissions, and hence the NH3 lifetime (e.g. Zhu et al., 2015). NH3 also contributes to the formation of atmospheric aerosols that can reside and be transported in the atmosphere for several days to a week releasing NH3 back into the atmosphere modifying the variability of ammonia concentrations. It would be good if the authors could provide insights on the impacts of the variability due to bi-directional flux.

Zhu L., D.K. Henze, J.O. Bash, G.-R. Jeong, K.E. Cady-Pereira, M.W. Shephard, M. Luo, F. Paulot, and S. Capps, Global Evaluation of Ammonia Bi-Directional Exchange and Livestock Diurnal Variation Schemes, Atmos. Chem. Phys, 15, 12823-12843, doi:10.5194/acp-15-12823-2015, 2015.

Thank you to the referee for this suggestion. We have added discussion of bi-directional flux and its implementation in Sec. 2.2. While we cannot comment on the variability of ammonia due to bi-directional flux to the same extent as to which we comment later on the variability due to acid-precursor emissions and meteorology, we can gather from Zhu et al., 2015 that the effect may be significant under conditions during which a sufficiently large ammonium pool can form and be re-emitted. Bi-directional flux will not, however, account for variations in ammonia concentrations so large that they offset changes required by the primary emissions.

We have added discussion beginning on page 5, line 9: "We have not included any scheme which accounts for the bidirectional flux (deposition and re-emission) of ammonia in our base scenario. Rather, ammonia is permanently removed via wet scavenging in convective and stratiform precipitation (Mari et al., 2000; Amos et al., 2012) and via surface resistance-driven dry deposition (Wesley, 1989). Ongoing research suggests that a unidirectional dry deposition scheme may be inappropriate with regards to ammonia (Massad et al., 2010; Zhang et al., 2010). Under a bidirectional scheme, ammonia can be either taken-up or re-emitted by a plant based on the comparison of the ambient ammonia concentration with a varying compensation point (an ambient concentration greater than the compensation point leads to deposition). Re-emitted ammonia has the potential to affect ecosystems farther downwind. Failing to account for this re-emission may locally cause an overestimation in dry deposition resulting in low ammonia concentrations. Zhu et al. (2015) incorporate the bidirectional flux scheme of Pleim et al. (2013) into GEOS-Chem, which increases the July ammonia emissions and concentration in the US. This slightly reduces the July model bias compared to measurements at Ammonia Monitoring Network (AMoN) sites. However, the bidirectional scheme causes a decrease in ammonia emissions and

concentration in April and October, which worsens the comparison with observations and does not account for missing primary emissions. Such bidirectional flux schemes, developed largely to simulate field conditions, require higher resolution observations for evaluation at finer scales than those offered by current observations and global models."

2) The paper recognizes the limitation of the satellite observations due to lack of vertical information. It would be good to note that this is not general to satellite observations, but the particular IASI ammonia retrieval algorithm used in the study. For example, the new CrIS NH3 optimal estimation retrievals (Shephard et al., 2015) will be able to provide this type of information (e.g. averaging kernels and covariance matrics) allowing for more quantitative comparisons against the model simulations.

We agree that the limitation is specific to this IASI product. Since the CrIS product is not currently available for our use, we have added a reference to this improved product in our conclusions when referring to future ammonia observation systems.

We added in page 22, line 27: "New satellite ammonia products (e.g. from CrIS) with dense observations may better provide observational constraints, allowing for a more quantitative comparison with models."

3) It is still not totally clear how the impact of the spatial sampling between the model and the observations impact the measurement variability. For a study over just North America, why was a global GEOS-Chem model used instead of a more regional model (i.e. CMAQ) to investigate the ammonia variability? A regional model would at least have a spatial sampling that is more representative for comparisons with the observations.

We use the nested grid resolution of GEOS-Chem, which provides a finer spatial resolution than a typical global model simulation. This study follows Schiferl et al. (2014), which was performed using the same model and has been applied extensively to ammonia studies in the US (see Sec. 2.3). While our study focuses on the US, in large part because there are more in situ observations over this region, we hope that our analysis in this study may inform the simulation of ammonia in other regions of the world simulated by GEOS-Chem.

Minor Comments:

1) Page 5, line 27. Also should add in AIRS and CrIS.

**Added references to page 6, line 14: "; Shephard and Cady-Pereira, 2015; Warner et al., 2015)"**

2) Page5, lines 29-30. ": : :calculated from a wider spectral range than previous ammonia products,: ::". It is not clear if the point is to just state this fact, or imply that this is better. Using

a wider spectral coverage does not necessarily produce a better retrieved product. For example, a robust spectral window selection approach can be based on the maximum information content by taking into consideration errors (e.g. interfering species, spectroscopic errors, measurement errors, etc.) (e.g. Echle et al., (2000) and Worden et al. (2004)).

Echle, G., T. von Clarmann, A. Dudhia, J. M. Flaud, B. Funke, N. Glatthor, B. Kerridge, M. Lopez-Puertas, F. J. Martin-Torres, and G. P. Stiller (2000), Optimized spectral microwindows for data analysis of the Michelson Interferometer for Passive Atmospheric Sounding on the environmental satellite, Appl. Opt., 39(30), 5531–5540.

Worden J, S. Sund, M.W Shephard, S.A Clough, H. Worden, K Bowman, A Goldman. Predicted errors of tropospheric emission spectrometer nadir retrievals from spectral window selection. J Geophys Res. 2004;109:doi:10.1029/2004JD004522.

Thank you to the referee for pointing out this need for clarification. The wider spectral range is mentioned to imply that it is better because it allows for more sensitivity to the weak ammonia signal. This is fully described in Sec. 3 of Van Damme et al. (2014a). We do acknowledge that this does not come without trade-offs, such as the loss of all vertical sensitivity information. We clarify this in the text.

In page 6, line 18, we added "to increase sensitivity".

3) Page 6, line 1: Please state what forward radiative transfer model was used.

**Done. Added to page 6, line 20: "the Atmosphit"**

4) Page 6, line 5: Are these uncertainties relative, or absolute, or both?

**As described in Van Damme et al. (2014a), these are relative. This is clarified in the text.**

**Added to page 6, line 23: "relative"**

5) Page 6, line 14: remove "present"

**Done. Removed "present", changed "attempts" to "attempt" in page 7, line 1.**

6) Page 6, line 16: Maybe also add to the line ending in ": : :distributed measurements" the additional "and the differences in measured quantities.", which leads nicely into the next sentence.

**Changed as suggested. Added "and the differences in measured quantities" to page 7, line 3.**

7) Page 9, lines 25-29: should mention in addition to vertical sensitivity, the last of the actual information content limits the comparison.

We are not clear to which other information the referee is referring. If the information (vertical sensitivity?) were available to treat the model output as the IASI measurements, the comparison would be exact. Therefore while the IASI retrieval may be imperfect, a perfect comparison should still be possible.

8) Page 10, lines 10-15. Could the lack of variability also be due to the fact that satellite total column values are being used, rather than information from only the parts of the profile where the satellite is sensitive (e.g. often limited information right at the surface).

We agree, and this is consistent with the previous discussion of the weaknesses relating to the missing vertical sensitivity information from the IASI product. We have added more explicit discussion of this in the text.

[revised manuscript text omitted]

- 5 standard model version) are extended uniformly spatially to 2011 and 2012 from EPA Trends data (www.epa.gov/ttn/chief/trends/). Mean anthropogenic SOx, largely from power generation, and NOx, largely from automobiles, emission rates over the US in summer (JJA) 2008 are 18 mg S km-2 s-1 and 16 mg N km-2 s-1, respectively. As shown in Fig. 1, anthropogenic SOx and NOx emissions are highest in the eastern US and are often associated with rural point or dense urban sources. These emission rates decrease by 30 % and 33 %, respectively, by 2012. The majority of the magnitude
- of these decreases occurs in the eastern regions of the US. For 2008, anthropogenic SOx makes up 98 % of total SOx emissions, and anthropogenic NOx makes up 65 % of total NOx emissions. Other major sources of NOx 
[revised manuscript text omitted]